# From Minor Adjustment to Major Gains: Soft Logit Normalization Loss Enhances Representations and Generalization

## Abstract

Developing novel loss functions for small models to attain performance parity with their larger counterparts is an active research area in artificial intelligence. We propose the Soft Logit Normalization (SLN) loss, which normalizes the logit vector by its powered L2-norm before applying the standard softmax function. In comparison with the classical cross-entropy loss, SLN loss significantly improves generalization across multiple vision benchmarks, including CIFAR-10 and ImageNet-1K, enabling small models to match the performance of models with approximately three times more parameters—an improvement comparable to that achieved by advanced knowledge distillation techniques. Beyond vision tasks, experiments on language tasks with large transformer-based models (e.g., BERT$_{LARGE}$ with 340M parameters) demonstrate the versatility of SLN loss across modalities. Theoretical analysis further show that SLN loss facilitates more separable penultimate-layer representations, which contributes to better generalization, as numerically validated on diverse datasets. This work not only advances the practical deployment of efficient models on resource-constrained devices but also opens new directions for research into loss function design.

## 1 Introduction

Due to the superior generalization performance when scaling from small to large models (Kaplan et al., 2020; Hoffmann et al., 2022), recent AI research has been dominated by large models (DeepSeek-AI, 2025; Zhao et al., 2023; Achiam et al., 2023; Liu et al., 2024). The success of knowledge distillation (Hinton et al., 2015; Gou et al., 2021; DeepSeek-AI, 2025)—a technique where the representations of a large teacher model guide the training of a smaller student model—demonstrates that large models inherently possess strong representations. Moreover, small models can achieve higher performance when provided with improved representations. However, knowledge distillation typically requires the training of a high-performing large model, which is computationally expensive. Therefore, understanding the principles of representation scaling from small to large models, and leveraging these principles to directly train small models with high-quality representations comparable to those of large models, warrants further investigation.

Through numerical experiments in classification tasks trained with cross-entropy (CE) loss, when considering the post-train logit vector is normalized by its L2-norm with varying power hyperparameters, we find that large models consistently increase logit difference. By incorporating the above principle into training, we propose the soft logit normalization (SLN) loss and examine its effect on enabling small models to achieve high-quality representations comparable to those of large models.

Evaluations on the image and language classification tasks demonstrate that the SLN loss significantly enhances generalization compared to CE loss, enabling small models to achieve performance comparable to larger models (e.g., ResNet-20 versus ResNet-56 and BERT$_{BASE}$ versus BERT$_{LARGE}$). Furthermore, the generalization improvements achieved with SLN loss are nearly equivalent to those obtained using advanced knowledge distillation techniques (Xu et al., 2020; Zhao et al., 2020). On the ImageNet-1K dataset (Deng et al., 2009), the SLN loss delivers substantial improvements in top-1 accuracy, comparable to those achieved with extensive data augmentation combined with regularization strategies.

To elucidate the mechanism underlying the SLN loss, we perform a theoretical analysis, which demonstrates that its gradient at the penultimate layer promotes more separable representations, thereby enhancing generalization. Numerical experiments on synthetic and CIFAR-10 datasets further validate our theoretical results.

This work not only provides a computationally efficient alternative to knowledge distillation for training high-performance compact models but also advances our theoretical understanding of representation learning in neural networks. Our findings potentially provide valuable insights into leveraging the underlying principles when scaling from small to large models.

## 2 RELATED WORK

**Knowledge distillation.** Recent studies have extensively explored various knowledge distillation methods, which can be broadly categorized into three types: logit-based, feature-based, and relation-based approaches (Gou et al., 2021). Logit-based methods utilize the outputs of the teacher model's final layer as guidance for training the student model, achieving promising results (Hinton et al., 2015; Meng et al., 2019; Lv et al., 2024). Feature-based methods, also referred to as representation learning (Bengio et al., 2013), leverage intermediate representations from the teacher model as knowledge to supervise the student model's training, thereby narrowing the performance gap between the teacher and student models (Jin et al., 2019; Passban et al., 2021). In contrast, relation-based methods focus on capturing relationships either between representations across different layers or among data samples, rather than relying on specific layer outputs. This approach enables the student model to learn inter-layer or inter-sample correlations, thereby improving generalization performance (Passalis et al., 2020; Chen et al., 2020).

**Designing new loss functions.** The direct design of novel loss functions has emerged as a promising alternative to knowledge distillation, attracting considerable attention in recent research. For instance, loss functions incorporating the cosine function applied to logits, with the goal of achieving a large margin, have demonstrated performance improvements in image classification tasks (Liu et al., 2016). Similarly, subtracting a constant from the logit corresponding to the true label has been proposed in Wang et al. (2018b) to increase the model's margin and enhance its generalization capabilities. Additionally, performing normalization technics to logit vector has been explored for various applications. For example, logit normalization loss has been introduced to mitigate neural network overconfidence in out-of-distribution detection (Wei et al., 2022) and to solve long-tail object detection (Zhao et al., 2024). Furthermore, comprehensive investigations into various novel loss functions, including logit normalization, have been conducted to analyze their relationships with transfer learning (Kornblith et al., 2021) and model calibration (Cattelan & Silva, 2023). In contrast, our SLN loss normalize the logit vector by its powered L2-norm and is capable of exhibiting superior generalization performance.

## 3 METHODS

We consider a supervised $k$-class classification task. Let $\{(\mathbf{x_n}, y_n)\}_{n=1}^{N}$ represent the trainable dataset, where $\mathbf{x_n} \in \mathbb{R}^d$ is the input and $y_n \in \{1, 2, \ldots, k\}$ is its corresponding label. We denote $\mathbf{f} : \mathbb{R}^d \to \mathbb{R}^k$ as a classifier (e.g., a neural network) with trainable parameters $\theta$, which maps an input to the output space. The output $\mathbf{f}(\mathbf{x_n}; \theta)$ is referred to as the logit or the pre-softmax output of $\mathbf{x_n}$.

**Cross-Entropy Loss.** The cross-entropy (CE) loss (Bridle, 1989; 1990) is a widely used objective function for classification tasks. It is defined as:

$$\mathcal{L}_{CE}(\mathbf{f}(\mathbf{x_n}; \theta)) = -\log \frac{e^{f_{y_n}(\mathbf{x_n}; \theta)}}{\sum_{i=1}^{k} e^{f_i(\mathbf{x_n}; \theta)}}, \tag{1}$$

where $f_i(\mathbf{x_n}; \theta)$ represents the $i$-th component of $\mathbf{f}(\mathbf{x_n}; \theta)$.

While training with CE loss, we find a consistent positive correlation between test accuracy and logit difference when the logit vector is normalized by its powered L2-norm (the definition of the latter one is: $= \sum_{n=1}^{N} \sum_{i \in [k] \backslash y_n} [f_{y_n}(\mathbf{x_n}; \theta) - f_i(\mathbf{x_n}; \theta)] / \|\mathbf{f}(\mathbf{x_n}; \theta)\|_2^\gamma$), as shown in Figure 1. This suggests

that increasing the logit difference under such normalization may effectively improve generalization. Motivated by this finding, we propose our SLN loss.

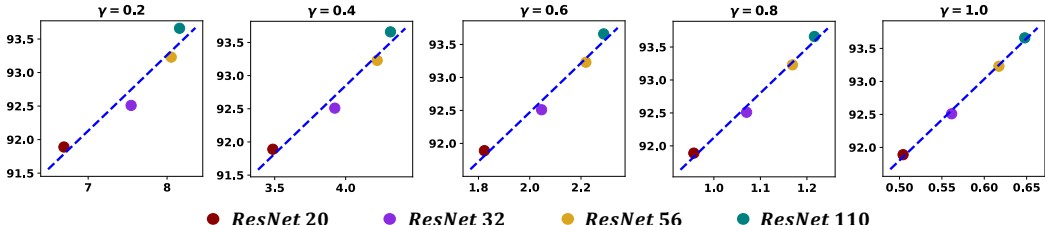

Figure 1: Relationship between test accuracy (y-axis) and logit difference (x-axis) with different $\gamma$. Training is conducted on CIFAR-10 using different ResNet architectures (He et al., 2016) with CE loss.

**Soft Logit Normalization Loss.** The soft logit normalization (SLN) loss is defined as:

$$\mathcal{L}_{SLN}(\mathbf{f}(\mathbf{x_n};\theta)) = -\log \frac{e^{f_{y_n}(\mathbf{x_n};\theta)/(\tau\|\mathbf{f}(\mathbf{x_n};\theta)\|_2^\gamma)}}{\sum_{i=1}^k e^{f_i(\mathbf{x_n};\theta)/(\tau\|\mathbf{f}(\mathbf{x_n};\theta)\|_2^\gamma)}}, \tag{2}$$

where $\gamma$ is a tunable hyperparameter and $\tau$ is the temperature coefficient. Notably, when $\gamma = 1$, the SLN loss reduces to the logit normalization loss studied in Wei et al. (2022), which is originally proposed for out-of-distribution detection. For clarity, we refer to this special case ($\gamma = 1$) as the logit normalization (LN) loss. For the hyperparameters $\gamma$ and $\tau$, we find that selecting $\gamma$ within the range $[0.6, 0.8]$ yields consistently good performance. Thus, unless otherwise specified, we set $\gamma = 0.7$ as the default. The optimal value of $\tau$ depends on the number of classes. Further details on the selection of $\gamma$ and $\tau$ are provided in Appendix A.

## 4 RESULTS

This section provides extensive evidence demonstrating that SLN loss enhances generalization across various task types (e.g., image and language classification tasks) and datasets of different scales (e.g., CIFAR-10, CIFAR-100, ImageNet-1K and GLUE). The implementation details can be found in Appendix B.

### 4.1 SLN LOSS ENABLES SMALL NETWORKS TO ACHIEVE LARGE NETWORKS GENERALIZATION

**Dataset and models.** Our initial experiments are conducted using the CIFAR dataset, which consists of 50,000 training images and 10,000 test images. The experimental setup adheres to the ResNet architectures described in He et al. (2016), including ResNet-20, ResNet-32, ResNet-56, and ResNet-110 models.

**Accuracy comparison with CE loss and LN loss.** Table 1 presents a comprehensive performance comparison across various loss functions. While the LN loss (Wei et al., 2022) performs similar to the standard CE loss, our proposed SLN loss demonstrates significant accuracy improvements on both the CIFAR-10 and CIFAR-100 datasets. Furthermore, we observe that SLN loss enables smaller models to achieve generalization performance comparable to networks approximately three times their size. For instance, a ResNet-20 (0.27M parameters) trained with SLN loss performs on par with a ResNet-56 (0.86M parameters) using CE loss. Similarly, a ResNet-32 (0.46M parameters) with SLN loss matches the performance of a ResNet-110 (1.73M parameters) with CE loss. These results underscore the effectiveness of the additional soft hyperparameter in SLN, which is the key distinction from LN loss and the primary driver of the observed performance gains.

**Accuracy comparison across different methods.** In comparison with other logit-based designed loss function (e.g., M-Softmax (Wang et al., 2018b) and L-Softmax (Liu et al., 2016)), our SLN loss

consistently achieves the highest test accuracy, thereby confirming its effectiveness. Additionally, when comparing the performance of ResNet-20 trained using knowledge distillation (Gou et al., 2021; Xu et al., 2020; Zhao et al., 2020) with our best-performing ResNet-20 model (92.70% accuracy on CIFAR-10 and 70.51% on CIFAR-100), it is evident that our method achieves comparable results without relying on larger teacher models.

Table 1: Comparative performance of ResNet models trained using different methods. The performance of ResNet-20 with knowledge distillation loss is obtained from the original paper (CIFAR-10: teacher model ResNet-56, 93.63%; CIFAR-100: teacher model ResNet-110, 71.65%). All other results are presented as the mean ± std across ten independent runs.

| Model | Loss | Acc. (CIFAR10) | Acc. (CIFAR100) |
|---|---|---|---|
| ResNet-20 | Cross-Entropy | 91.81±0.15% | 67.26±0.18% |
| | Logit Normalization | 92.05±0.10% | 67.33±0.16% |
| | L-Softmax | 92.11±0.15% | 67.80±0.16% |
| | M-Softmax | 92.21±0.14% | 67.78±0.15% |
| | Knowledge Distillation | **92.67%** | **70.75%** |
| | Soft Logit Normalization | 92.58±0.08% | 69.75±0.25% |
| ResNet-32 | Cross-Entropy | 92.40±0.24% | 68.62±0.33% |
| | Logit Normalization | 92.45±0.22% | 68.92±0.42% |
| | L-Softmax | 92.78±0.14% | 69.11±0.15% |
| | M-Softmax | 92.76±0.15% | 69.22±0.33% |
| | Soft Logit Normalization | **93.19±0.17%** | **71.41±0.21%** |
| ResNet-56 | Cross-Entropy | 93.07±0.17% | 70.19±0.22% |
| | Logit Normalization | 93.16±0.21% | 70.32±0.19% |
| | L-Softmax | 93.50±0.14% | 70.82±0.15% |
| | M-Softmax | 93.47±0.14% | 70.71±0.36% |
| | Soft Logit Normalization | **93.78±0.18%** | **72.76±0.23%** |
| ResNet-110 | Cross-Entropy | 93.64±0.16% | 70.84±0.31% |
| | Logit Normalization | 93.67±0.20% | 71.03±0.20% |
| | L-Softmax | 93.71±0.14% | 71.66±0.15% |
| | M-Softmax | 93.80±0.16% | 71.47±0.52% |
| | Soft Logit Normalization | **94.10±0.09%** | **73.64±0.29%** |

## 4.2 SLN LOSS FACILITATES GENERALIZATION COMPARABLE TO EXTENSIVE DATA AUGMENTATION

**Dataset and models.** We extend our experiments to the ImageNet-1K dataset Deng et al. (2009), which consists of approximately 1.28 million training images and 50,000 validation images distributed in 1,000 categories. To validate the effectiveness of the SLN loss on a larger-scale dataset and to characterize its generalization improvements, we employ the ResNet-50 architecture, a widely studied model in previous literature (He et al., 2016; Kornblith et al., 2021; Wightman et al., 2021; Touvron et al., 2019; Pytorch; Touvron et al., 2021).

**Training methods.** Training is conducted following the procedure described in Touvron et al. (2021); Wightman et al. (2021). We consider two distinct data augmentation and regularization strategies: The first strategy utilizes basic augmentation techniques, including RandAugment (Cubuk et al., 2020) and random erasing (Zhong et al., 2020). The second strategy adopts a more comprehensive approach, incorporating additional advanced techniques such as mixup (Zhang et al., 2017), cutmix (Yun et al., 2019) and label smoothing (Szegedy et al., 2016), as implemented in the *timm* library (Wightman, 2019).

**Accuracy comparison.** Table 2 presents the performance results on the ImageNet-1K dataset. It shows that ResNet-50 (26M parameters) trained with our proposed SLN loss, even without extensive advanced data augmentation and regularization techniques, achieves performance comparable not

only to larger models trained with CE loss (e.g., ResNet-101 (45M parameters)), but also to the original ResNet-50 trained with CE loss using comprehensive data augmentation and regularization strategies. These results indicate that our SLN loss effectively facilitates generalization performance similar to that achieved through extensive data augmentation.

Table 2: Comparative performance of ResNet models trained on ImageNet-1K.

| Model | Loss | With advanced data augmentation | Top-1 acc. |
|---|---|---|---|
| ResNet-50 | Cross-Entropy | - | 76.6% |
| | Cross-Entropy | ✓ | 78.2% |
| | Soft Logit Normalization | - | 78.0% |
| ResNet-101 | Cross-Entropy | - | 77.9% |

### 4.3 SLN LOSS ENHANCES GENERALIZATION ON LANGUAGE TASK

**Dataset and models.** To evaluate our method on language tasks, we fine-tune $\text{BERT}_{BASE}$ (110M parameters) and $\text{BERT}_{LARGE}$ (340M parameters) on the General Language Understanding Evaluation (GLUE) benchmark (Wang et al., 2018a), following the protocol of Devlin et al. (2019). Our primary objective is to assess whether the generalization benefits of the SLN loss extend to language tasks and Transformer architectures (Vaswani et al., 2017).

Table 3: GLUE Evaluation results. The number below each task denotes the number of training examples. F1 scores are reported for QQP and MRPC, Matthew's correlations are reported for CoLA, and Accuracy scores are reported for other tasks.

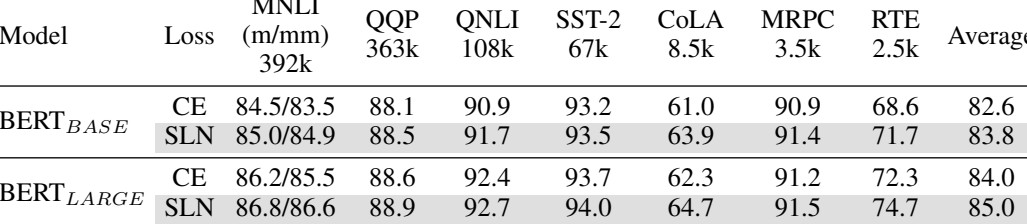

| Model | Loss | MNLI (m/mm) 392k | QQP 363k | QNLI 108k | SST-2 67k | CoLA 8.5k | MRPC 3.5k | RTE 2.5k | Average |
|---|---|---|---|---|---|---|---|---|---|
| $\text{BERT}_{BASE}$ | CE | 84.5/83.5 | 88.1 | 90.9 | 93.2 | 61.0 | 90.9 | 68.6 | 82.6 |
| | SLN | 85.0/84.9 | 88.5 | 91.7 | 93.5 | 63.9 | 91.4 | 71.7 | 83.8 |
| $\text{BERT}_{LARGE}$ | CE | 86.2/85.5 | 88.6 | 92.4 | 93.7 | 62.3 | 91.2 | 72.3 | 84.0 |
| | SLN | 86.8/86.6 | 88.9 | 92.7 | 94.0 | 64.7 | 91.5 | 74.7 | 85.0 |

**Performance comparison.** Table 3 presents the GLUE results on the evaluation sets. Similar to the results shown in Table 1, these findings demonstrate that the SLN loss enables smaller models ($\text{BERT}_{BASE}$ with 110M parameters) to achieve generalization performance comparable to that of networks approximately three times their size ($\text{BERT}_{LARGE}$ with 340M parameters) trained with CE loss. This indicates that our conclusions can be extended to language-based tasks and Transformer architectures. Another interesting observation concerns the performance on MNLI: for models trained with CE loss, the accuracy on the mismatch category is lower than on the match category, which is expected; however, training with SLN loss appears to reduce this gap, resulting in similar accuracies for both categories.

### 4.4 ABLATION STUDIES

To demonstrate that the performance improvements presented in the preceding sections are primarily attributable to the proposed soft hyperparameter $\gamma$ in the SLN loss, we conducted ablation studies on CIFAR-10 using ResNet-20. The results are summarized in the table below.

From these results, we observe that the generalization gains of SLN are primarily attributed to the parameter $\gamma$. However, the temperature parameter $\tau$ also plays a crucial role in SLN: while lowering $\tau$ is generally detrimental for CE loss, it is beneficial for SLN, further improving its performance. This suggests that the combination of soft hyperparameter ($\gamma$) and appropriate temperature scaling ($\tau$) is key to SLN's effectiveness.

Table 4: Ablation study on the impact of different training losses and hyperparameters. All results are reported as mean $\pm$ standard deviation across ten independent runs. Here $\gamma = 0.7$ for SLN loss.

| $\tau$ | 0.1 | 0.2 | 0.4 | 0.8 | 1.0 |
|---|---|---|---|---|---|
| CE | 90.78±0.28% | 91.11±0.25% | 91.34±0.21% | 91.77±0.23% | **91.81±0.15%** |
| LN | 91.85±0.13% | **92.05±0.10%** | 91.94±0.15% | 91.78±0.09% | 91.52±0.33% |
| SLN | 92.30±0.12% | **92.58±0.07%** | 92.24±0.13% | 92.19±0.22% | 91.77±0.12% |

## 5 THEORETICAL ANALYSIS

In this section, we analyze how the proposed SLN loss enhances generalization. To theoretically investigate the underlying mechanism of the SLN loss, we consider a model comprising: (1) a linear classifier with trainable weights (representing the final layer), and (2) trainable data points (corresponding to penultimate-layer representations). The complete theoretical derivations are provided in Appendix C.

### 5.1 BINARY CLASSIFICATION

#### 5.1.1 TRAINABLE DATA POINTS

Let $\{(\mathbf{x_n}, y_n)\}_{n=1}^{N}$ represent the trainable data points, where $\mathbf{x_n} \in \mathbb{R}^d$ and $y_n \in \{1, 2\}$. The linear classifier is defined as $f_1(\mathbf{x}) = \mathbf{w_1} \cdot \mathbf{x} + b_1$ and $f_2(\mathbf{x}) = \mathbf{w_2} \cdot \mathbf{x} + b_2$, where $\mathbf{w_1}, \mathbf{w_2} \in \mathbb{R}^d$ and $b_1, b_2 \in \mathbb{R}$ are weights. Additionally, let $C_1 = \{n \mid y_n = 1\}$ and $C_2 = \{n \mid y_n = 2\}$ denote the index sets for classes 1 and 2, respectively. Under this setup, we can compute the gradients of the data points under different loss functions, which can also be interpreted as the gradient field in the representation space.

**Proposition 1.** *(Cross-Entropy) The gradients of the data points are given by:*

$$\frac{\partial \mathcal{L}_{CE}}{\partial \mathbf{x_n}} = \frac{e^{f_2(\mathbf{x_n})}}{e^{f_1(\mathbf{x_n})} + e^{f_2(\mathbf{x_n})}}(\mathbf{w_2} - \mathbf{w_1}), \ \forall n \in C_1,$$

$$\frac{\partial \mathcal{L}_{CE}}{\partial \mathbf{x_n}} = \frac{e^{f_1(\mathbf{x_n})}}{e^{f_1(\mathbf{x_n})} + e^{f_2(\mathbf{x_n})}}(\mathbf{w_1} - \mathbf{w_2}), \ \forall n \in C_2.$$

Proposition 1 provides an analytical expression for the gradients of the data points. Notably, under the gradient descent algorithm for optimizing the CE loss, the direction of the gradients is perpendicular to the classification hyperplane ($\mathbf{w_1} \cdot \mathbf{x} + b_1 = \mathbf{w_2} \cdot \mathbf{x} + b_2$). Additionally, the gradient magnitudes for different data points depend on the values of $f_2(\mathbf{x_n})$ and $f_1(\mathbf{x_n})$ for class 1 and class 2 data points, respectively. This indicates that data points misclassified or located closer to the classification hyperplane exhibit larger gradient magnitudes compared to well-classified points. Consequently, this results in compression along the direction aligned with the normal vector of the classification hyperplane.

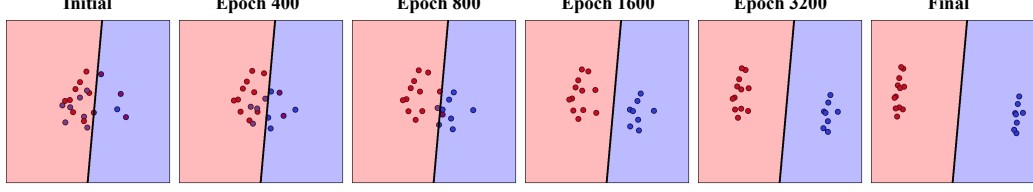

Figure 2: Training dynamics ($d = 2$) with CE loss. Data points are randomly sampled at initialization, with the weights of the linear classifier fixed while only the data points are trainable. Training is performed using gradient descent with a learning rate of 0.1 over 6000 epochs.

Figure 2 numerically validates this analysis, showing that, at the final stage of training, data points within the same class converge to lines almost parallel to the classification boundary, while maintaining their distribution along that direction as in the initialization.

We now consider the LN loss. Specifically, we can prove the following proposition:

**Proposition 2.** *(Logit Normalization) Let* $\Omega = \{\mathbf{x} \in \mathbb{R}^d \mid f_1(\mathbf{x}) = f_2(\mathbf{x}) = 0\}$. *Then, for all* $\mathbf{x_n} \notin \Omega$ *and* $\mathbf{x} \in \Omega$, *the following equation holds:*

$$\langle \frac{\partial \mathcal{L}_{LN}}{\partial \mathbf{x_n}}, \mathbf{x_n} - \mathbf{x} \rangle = 0, \tag{3}$$

*and if assume* $\mathbf{w_1}, \mathbf{w_2}$ *are linearly independent, non-zero vectors, then* $\frac{\partial \mathcal{L}_{LN}}{\partial \mathbf{x_n}} = 0$ *if and only if* $f_1(\mathbf{x_n}) + f_2(\mathbf{x_n}) = 0$.

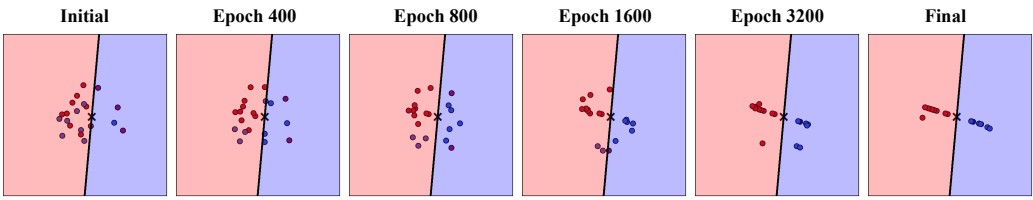

Figure 3: Training dynamics ($d = 2$) using the LN loss (SLN loss with $\gamma = 1$). All other settings are identical to those in Figure 2. The symbol "×" denotes the circle center $\mathbf{x}^*$ (i.e., $\Omega$).

Proposition 2 provides insight into the direction of the gradients of the data points. Consider the special case where $d = 2$, and assume $\mathbf{w_1}, \mathbf{w_2} \neq \mathbf{0}$ with $\mathbf{w_1} \neq \mathbf{w_2}$. In this scenario, $\Omega$ contains only a single point, say $\Omega = \{\mathbf{x}^*\}$. Equation (3) then reduces to:

$$\langle \frac{\partial \mathcal{L}_{LN}}{\partial \mathbf{x_n}}, \mathbf{x_n} - \mathbf{x}^* \rangle = 0,$$

which implies that the trajectory of $\mathbf{x_n}$ under gradient descent for optimizing this loss forms a circle centered at $\mathbf{x}^*$. Furthermore, if $\|\mathbf{w_1}\|_2 = \|\mathbf{w_2}\|_2$, the trajectory of $\mathbf{x_n}$ converges to a state where $\mathbf{x_n} - \mathbf{x}^*$ is perpendicular to the classification boundary. Figure 3 numerically shows that data points converge to the line $f_1(\mathbf{x}) + f_2(\mathbf{x}) = 0$, which is nearly perpendicular to the classification boundary. This behavior varies with the case trained using CE loss. Additionally, when $d > 2$, it is straightforward to deduce that the trajectory will form a circular curve on the surface of a cylinder.

Finally, we consider the general case of SLN loss and prove the following theorem:

**Theorem 1.** *(Soft Logit Normalization) For all* $\mathbf{x_n} \notin \Omega$ ($\Omega$ *is defined the same as in proposition 2), the following equation holds:*

$$\frac{\partial \mathcal{L}_{SLN}}{\partial \mathbf{x_n}} = c_1 \gamma \frac{\partial \mathcal{L}_{LN}}{\partial \mathbf{x_n}} + c_2 (1 - \gamma) \frac{\partial \mathcal{L}_{CE}}{\partial \mathbf{x_n}}, \tag{4}$$

*where* $c_1$ *and* $c_2$ *are real positive numbers that depend on* $\gamma, \tau, \mathbf{f}(\mathbf{x_n})$.

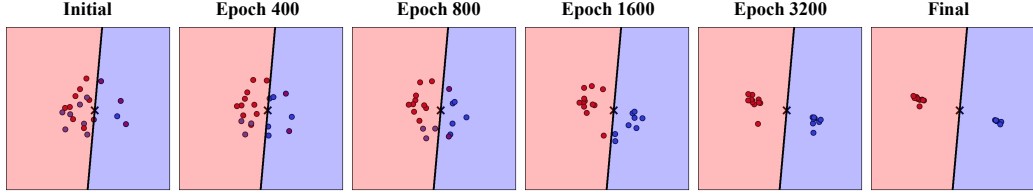

Figure 4: Training dynamics ($d = 2$) using the SLN loss with $\gamma = 0.7$. All other settings are identical to those in Figure 2. The symbol "×" denotes the circle center $\mathbf{x}^*$ (i.e., $\Omega$) presented in Proposition 2.

Theorem 1 indicates that the direction of $\frac{\partial \mathcal{L}_{SLN}}{\partial \mathbf{x_n}}$ is a linear combination of $\frac{\partial \mathcal{L}_{LN}}{\partial \mathbf{x_n}}$ and $\frac{\partial \mathcal{L}_{CE}}{\partial \mathbf{x_n}}$. This implies that SLN loss introduces a compression effect in different directions, resulting in greater within-class compression of data points. This behavior is demonstrated in the numerical experiment shown in Figure 4.

### 5.1.2 TRAINABLE WEIGHTS AND DATA POINTS

The above phenomenon can also be observed when the weights of the linear classifier are trainable. Figure 5 shows the results when both the weights and data points are trainable. It can be observed that our proposed SLN loss ($\gamma = 0.7$) consistently achieves the greatest within-class compression of data points, whereas both the CE loss and the LN loss only compress data points along a single direction.

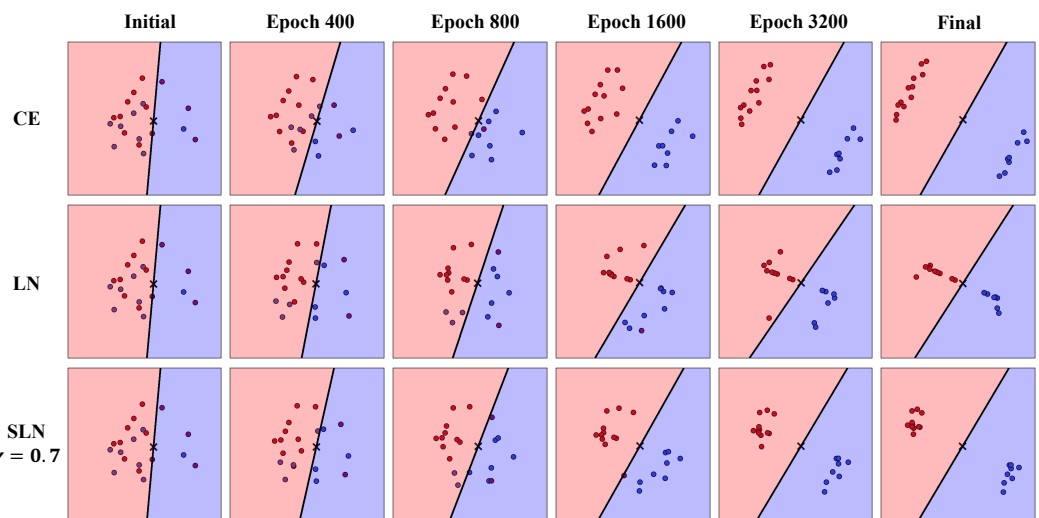

Figure 5: Training dynamics ($d = 2$) using different loss functions. The learning rate for the weights is set to $0.1/N$ to ensure that the changes in the weights and data points are of the same order with respect to $N$, where $N$ is the number of data points. All other settings are identical to those in Figure 2. The symbol "$\times$" denotes the circle center $\mathbf{x}^*$ (i.e., $\Omega$) presented in Proposition 2.

### 5.2 GENERAL CASE

The binary-case analysis naturally extends to the multi-class setting. Proofs of the multi-class versions of Proposition 1, Proposition 2, and Theorem 1 are provided in Appendix C.2. Consistent with the binary case, the theorem shows that SLN loss induces more compression directions than CE loss, strengthening within-class compression and improving representation capacity. Appendix C.3 further discusses how the analysis informs the training dynamics of large neural networks.

We validate the theory with experiments on CIFAR-10 using ResNet-20 and ResNet-32 trained under CE, LN, and SLN losses. To quantify representation quality, we compute the intra-inter ratio of penultimate-layer features:

$$\text{intra-inter ratio} = \frac{\text{mean}(\|x_{n_i} - c_i\|_2)}{\text{mean}(\|c_i - c_j\|_2)},$$

where $x_{n_i}$ is a penultimate-layer feature from class $i$ and $c_i = \text{mean}(x_{n_i})$ is its class center.

Figure 6 shows that SLN consistently yields the lowest intra-inter ratio, confirming the theory and highlighting that SLN is not a simple interpolation of CE and LN but a distinct dynamic regime. Notably, ResNet-20 with SLN achieves a lower ratio than ResNet-32 with CE or LN, as also visible in the t-SNE plots. This observation aligns with the test accuracies in Table 1, where ResNet-20 with SLN outperforms ResNet-32 trained with CE or LN, suggesting that penultimate-layer representations explains the performance gains.

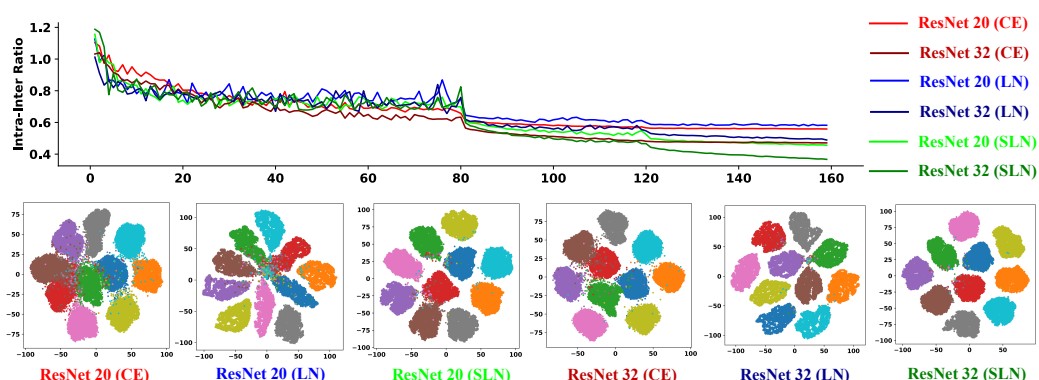

Figure 6: Top: Intra-inter ratio dynamics on CIFAR-10. Bottom: t-SNE visualization (Van der Maaten & Hinton, 2008) of penultimate-layer features at Epoch 160. Colors denote classes.

## 6 CONCLUSION

In this paper, we introduce the SLN loss, a novel loss function that significantly outperforms the traditional CE loss in terms of generalization across a diverse range of tasks. Through empirical evaluations, we demonstrate that the SLN loss enables small models to achieve performance comparable to larger models, which is typically achieved by knowledge distillation techniques. For instance, on the CIFAR-10 and GLUE datasets, small models such as ResNet-20 and BERT$_{BASE}$ trained with the SLN loss achieve generalization on par with larger models like ResNet-56 and BERT$_{LARGE}$. Moreover, on the ImageNet-1K dataset, the SLN loss delivers substantial improvements in top-1 accuracy, rivaling the results obtained with advanced data augmentation and regularization strategies. Furthermore, our theoretical analysis shows that the SLN loss promotes more separable representations compared to both CE and LN losses. Numerical experiments corroborate this finding by demonstrating that SLN induces a distinct training dynamics—rather than an intermediate state between CE and LN—thereby enhancing representation separability and overall performance.

Our study suggests potential applications of the SLN loss in other tasks, such as next-token prediction (Bengio et al., 2003). Additionally, our findings offer valuable insights for designing novel loss functions that enable small models to achieve performance comparable to larger models. We hope that this work contributes to the development of efficient, high-performing models suitable for resource-constrained devices.

### AUTHOR CONTRIBUTIONS

Our main contributions are as follows:

- We propose the SLN loss, a novel loss function that introduces an additional power hyperparameter $\gamma$ beyond the LN loss.

- Extensive experiments across diverse vision and language classification tasks show that SLN consistently outperforms both CE and LN losses, demonstrating its practical effectiveness. An ablation study further highlights the critical role of the hyperparameter $\gamma$ in achieving performance gains.

- We establish theoretical insights into the optimization behavior of SLN, revealing that it promotes greater representation separability than CE and LN. Empirical results verify this finding by demonstrating that SLN operates under a distinct training regime, which accounts for its enhancing representation separability and consistent performance improvements.

### REPRODUCIBILITY STATEMENT

The main text, together with Appendices A and B, provides comprehensive details for reproducing our experiments, including hyperparameters, data augmentation strategies, and other implementation

settings. The code will be released upon reasonable request. For the theoretical results, Section 5 and Appendix C present the full set of assumptions and complete proofs.

LLM USAGE

Large Language Models (LLMs) were used to improve the writing throughout the text and to find related work in Section 2.

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

## A  CHOICE OF HYPERPARAMETERS IN SLN LOSS

**Temperature coefficient $\tau$.** Our findings indicate that the optimal value of $\tau$ depends on the number of classes. Specifically, assuming there are $k$ classes, we observe that $\|\mathbf{f}(\mathbf{x};\theta)\|_2^\gamma = O(k^{\frac{\gamma}{2}})$ with random initialization. To ensure that $f_{y_n}(\mathbf{x};\theta)/(\tau\|\mathbf{f}(\mathbf{x};\theta)\|_2^\gamma)$ remains within a proper range for stabilizing the training process, we require $\tau = O(k^{-\frac{\gamma}{2}})$. This analysis is supported by the performance results presented in Table 5.

Table 5: Comparative performance for different choices of $\tau$ on CIFAR-10 ($k = 10$), CIFAR-100 ($k = 100$), and ImageNet-1K ($k = 1000$), with $\gamma = 0.7$.

| $\tau$ | CIFAR-10 | CIFAR-100 | ImageNet-1K |
|---|---|---|---|
| 0.2 | **92.58±0.08%** | 67.68±0.33% | 76.1% |
| $0.08(\approx 0.2 * 10^{-0.35})$ | 91.78±0.25% | **69.75±0.25%** | 77.0% |
| $0.04(\approx 0.2 * 100^{-0.35})$ | 89.56±0.63% | 68.66±0.43% | **78.0%** |

**Soft hyperparameter $\gamma$.** The hyperparameter $\gamma$ determines the balance between making the SLN loss behave more like LN loss or CE loss. We conducted a grid search over $\gamma$ values in $\{0.0, 0.1, 0.2, \ldots, 0.9, 1.0\}$, as shown in Figure 7. The results show that $\gamma = 0.7$ yields the best performance, while values in the range $[0.6, 0.8]$ consistently yields strong performance.

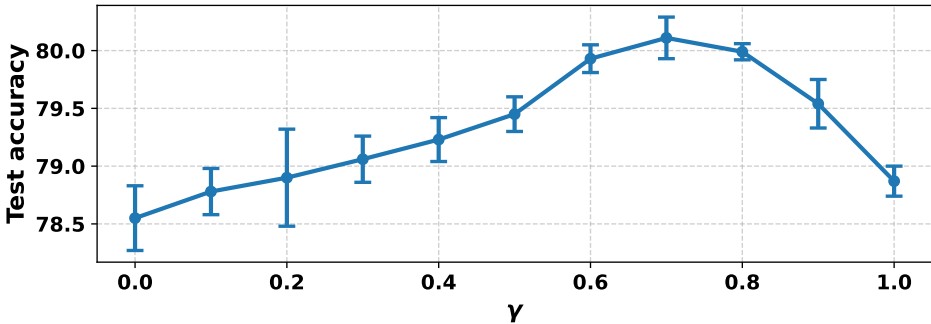

Figure 7: Mean test accuracy on CIFAR-10, CIFAR-100 and ImageNet-1K using SLN loss with different $\gamma$ values (with the optimal temperature coefficient $\tau$ as discussed above).

## B  EXPERIMENTAL SETTINGS

### B.1  CIFAR-10 AND CIFAR-100

The training process and data augmentation procedures follow the original protocol outlined in He et al. (2016), with the exception of the weight decay and learning rate settings. This adjustment is necessary because the optimal weight decay and learning rate differ between CE loss and SLN loss. The results reported in Table 1 represent the best performance for each method, selected from an initial learning rate in $\{0.01, 0.05, 0.1\}$ and a weight decay in $\{0.0001, 0.001, 0.005\}$. Specifically, the optimal settings for CE loss are learning rate $= 0.1$ and weight decay $= 0.0001$, while for SLN loss, the optimal settings are learning rate $= 0.01$ and weight decay $= 0.005$.

### B.2  IMAGENET-1K

The training settings for ResNet models on ImageNet-1K are summarized in Table 6. The primary differences in the training configurations lie in the use of advanced data augmentation and regularization techniques, including mixup Zhang et al. (2017), cutmix Yun et al. (2019) and label smoothing Szegedy et al. (2016).

Table 6: ImageNet-1K training settings in Table 2.

| Training config | CE | CE with advanced data augmentations | SLN |
|---|---|---|---|
| Optimizer | AdamW | AdamW | AdamW |
| Base learning rate | 1e-3 | 1e-3 | 3e-3 |
| Weight decay | 0.05 | 0.05 | 0.06 |
| Batch size | 1024 | 1024 | 1024 |
| Training epochs | 300 | 300 | 300 |
| Learning rate schedule | cosine | cosine | cosine |
| Warmup epochs | 5 | 5 | 5 |
| Warmup schedule | linear | linear | linear |
| Stochastic depth | 0.1 | 0.1 | 0.1 |
| RandAugment | rand-m9-mstd0.5 | rand-m9-mstd0.5 | rand-m9-mstd0.5 |
| Random erasing | 0.25 | 0.25 | 0.25 |
| Mixup | - | 0.8 | - |
| Cutmix | - | 1.0 | - |
| Label smoothing | - | 0.1 | - |

### B.3 GLUE

Training strictly follows the protocol of Devlin et al. (2019): we use a batch size of 32 and fine-tune for 3 epochs on all GLUE tasks. For each task, we report the best performance across learning rates $\{5\text{e-}5, 4\text{e-}5, 3\text{e-}5, 2\text{e-}5\}$.

## C   PROOFS OF THEOREMS

### C.1   BINARY CLASSIFICATION

Let $\{(\mathbf{x_n}, y_n)\}_{n=1}^N$ represent the trainable data points, where $\mathbf{x_n} \in \mathbb{R}^d$ and $y_n \in \{1, 2\}$. The linear classifier is defined as $f_1(\mathbf{x}) = \mathbf{w_1} \cdot \mathbf{x} + b_1$ and $f_2(\mathbf{x}) = \mathbf{w_2} \cdot \mathbf{x} + b_2$, where $\mathbf{w_1}, \mathbf{w_2} \in \mathbb{R}^d$ and $b_1, b_2 \in \mathbb{R}$ are weights. Additionally, let $C_1 = \{n \mid y_n = 1\}$ and $C_2 = \{n \mid y_n = 2\}$ denote the index sets for classes 1 and 2, respectively.

**Proposition.** *(Cross-Entropy) The gradients of the data points are given by:*

$$\frac{\partial \mathcal{L}_{CE}}{\partial \mathbf{x_n}} = \frac{e^{f_2(\mathbf{x_n})}}{e^{f_1(\mathbf{x_n})} + e^{f_2(\mathbf{x_n})}}(\mathbf{w_2} - \mathbf{w_1}), \ \forall n \in C_1,$$

$$\frac{\partial \mathcal{L}_{CE}}{\partial \mathbf{x_n}} = \frac{e^{f_1(\mathbf{x_n})}}{e^{f_1(\mathbf{x_n})} + e^{f_2(\mathbf{x_n})}}(\mathbf{w_1} - \mathbf{w_2}), \ \forall n \in C_2.$$

*Proof.* The cross-entropy loss can be expressed as:

$$\mathcal{L}_{CE}(\mathbf{x_n}) = \log(1 + e^{f_2(\mathbf{x_n}) - f_1(\mathbf{x_n})}), \ \forall n \in C_1,$$

$$\mathcal{L}_{CE}(\mathbf{x_n}) = \log(1 + e^{f_1(\mathbf{x_n}) - f_2(\mathbf{x_n})}), \ \forall n \in C_2.$$

Using the definitions $f_1(\mathbf{x_n}) = \mathbf{w_1} \cdot \mathbf{x_n} + b_1$ and $f_2(\mathbf{x_n}) = \mathbf{w_2} \cdot \mathbf{x_n} + b_2$, the result follows directly. $\qquad\square$

**Proposition.** *($\gamma = 1$, termed as Logit Normalization) Let $\Omega = \{\mathbf{x} \in \mathbb{R}^d \mid f_1(\mathbf{x}) = f_2(\mathbf{x}) = 0\}$. Then, for all $\mathbf{x_n} \notin \Omega$ and $\mathbf{x} \in \Omega$, the following equation holds:*

$$\langle \frac{\partial \mathcal{L}_{LN}}{\partial \mathbf{x_n}}, \mathbf{x_n} - \mathbf{x} \rangle = 0,$$

*and if assume $\mathbf{w_1}, \mathbf{w_2}$ are linearly independent, non-zero vectors, then $\frac{\partial \mathcal{L}_{LN}}{\partial \mathbf{x_n}} = 0$ if and only if $f_1(\mathbf{x_n}) + f_2(\mathbf{x_n}) = 0$.*

*Proof.* Define the following terms:

$$s(\mathbf{x_n}) = [f_1(\mathbf{x_n}) - f_2(\mathbf{x_n})]/(\tau \|\mathbf{f}(\mathbf{x_n})\|_2),$$

$$p_1(\mathbf{x_n}) = \frac{e^{f_1(\mathbf{x_n})/(\tau \|\mathbf{f}(\mathbf{x_n})\|_2)}}{e^{f_1(\mathbf{x_n})/(\tau \|\mathbf{f}(\mathbf{x_n})\|_2)} + e^{f_2(\mathbf{x_n})/(\tau \|\mathbf{f}(\mathbf{x_n})\|_2)}},$$

$$p_2(\mathbf{x_n}) = \frac{e^{f_2(\mathbf{x_n})/(\tau \|\mathbf{f}(\mathbf{x_n})\|_2)}}{e^{f_1(\mathbf{x_n})/(\tau \|\mathbf{f}(\mathbf{x_n})\|_2)} + e^{f_2(\mathbf{x_n})/(\tau \|\mathbf{f}(\mathbf{x_n})\|_2)}}.$$

The logit normalization loss can then be expressed as follows. Taking $n \in C_1$ as an example (the case for $n \in C_2$ is analogous):

$$\mathcal{L}_{LN}(\mathbf{x_n}) = \log(1 + e^{-s(\mathbf{x_n})}).$$

Using the chain rule of differentiation, we have:

$$\frac{\partial \mathcal{L}_{LN}}{\partial \mathbf{x_n}} = -p_2(\mathbf{x_n})\left(\frac{\partial s(\mathbf{x_n})}{\partial f_1(\mathbf{x_n})} \cdot \mathbf{w_1} + \frac{\partial s(\mathbf{x_n})}{\partial f_2(\mathbf{x_n})} \cdot \mathbf{w_2}\right).$$

After computation, the partial derivatives of $s(\mathbf{x_n})$ with respect to $f_1(\mathbf{x_n})$ and $f_2(\mathbf{x_n})$ are given by:

$$\frac{\partial s(\mathbf{x_n})}{\partial f_1(\mathbf{x_n})} = \frac{f_2^2(\mathbf{x_n}) + f_1(\mathbf{x_n})f_2(\mathbf{x_n})}{\tau \|\mathbf{f}(\mathbf{x_n})\|_2^3},$$

$$\frac{\partial s(\mathbf{x_n})}{\partial f_2(\mathbf{x_n})} = -\frac{f_1^2(\mathbf{x_n}) + f_1(\mathbf{x_n})f_2(\mathbf{x_n})}{\tau \|\mathbf{f}(\mathbf{x_n})\|_2^3}.$$

Observing these expressions, we find that:

$$f_1(\mathbf{x_n}) \cdot \frac{\partial s(\mathbf{x_n})}{\partial f_1(\mathbf{x_n})} + f_2(\mathbf{x_n}) \cdot \frac{\partial s(\mathbf{x_n})}{\partial f_2(\mathbf{x_n})} = 0.$$

Now, consider the inner product $\langle \frac{\partial \mathcal{L}_{LN}}{\partial \mathbf{x_n}}, \mathbf{x_n} \rangle$:

$$\langle \frac{\partial \mathcal{L}_{LN}}{\partial \mathbf{x_n}}, \mathbf{x_n} \rangle = -p_2(\mathbf{x_n})(\frac{\partial s(\mathbf{x_n})}{\partial f_1(\mathbf{x_n})} \cdot \langle \mathbf{w_1}, \mathbf{x_n} \rangle + \frac{\partial s(\mathbf{x_n})}{\partial f_2(\mathbf{x_n})} \cdot \langle \mathbf{w_2}, \mathbf{x_n} \rangle)$$

$$= -p_2(\mathbf{x_n})[\frac{\partial s(\mathbf{x_n})}{\partial f_1(\mathbf{x_n})}(f_1(\mathbf{x_n}) - b_1) + \frac{\partial s(\mathbf{x_n})}{\partial f_2(\mathbf{x_n})}(f_2(\mathbf{x_n}) - b_2)]$$

$$= p_2(\mathbf{x_n})(\frac{\partial s(\mathbf{x_n})}{\partial f_1(\mathbf{x_n})}b_1 + \frac{\partial s(\mathbf{x_n})}{\partial f_2(\mathbf{x_n})}b_2).$$

And for any $\mathbf{x} \in \Omega$, we similarly find:

$$\langle \frac{\partial \mathcal{L}_{LN}}{\partial \mathbf{x_n}}, \mathbf{x} \rangle = -p_2(\mathbf{x_n})(\frac{\partial s(\mathbf{x_n})}{\partial f_1(\mathbf{x_n})} \cdot \langle \mathbf{w_1}, \mathbf{x} \rangle + \frac{\partial s(\mathbf{x_n})}{\partial f_2(\mathbf{x_n})} \cdot \langle \mathbf{w_2}, \mathbf{x} \rangle)$$

$$= -p_2(\mathbf{x_n})[\frac{\partial s(\mathbf{x_n})}{\partial f_1(\mathbf{x_n})}(-b_1) + \frac{\partial s(\mathbf{x_n})}{\partial f_2(\mathbf{x_n})}(-b_2)]$$

$$= p_2(\mathbf{x_n})(\frac{\partial s(\mathbf{x_n})}{\partial f_1(\mathbf{x_n})}b_1 + \frac{\partial s(\mathbf{x_n})}{\partial f_2(\mathbf{x_n})}b_2).$$

Thus, we conclude that:

$$\langle \frac{\partial \mathcal{L}_{LN}}{\partial \mathbf{x_n}}, \mathbf{x_n} \rangle = \langle \frac{\partial \mathcal{L}_{LN}}{\partial \mathbf{x_n}}, \mathbf{x} \rangle.$$

Furthermore, since $p_2(\mathbf{x_n}) > 0$ and $\mathbf{w_1}, \mathbf{w_2}$ are linearly independent, non-zero vectors, $\frac{\partial \mathcal{L}_{LN}}{\partial \mathbf{x_n}} = 0$ is equivalent to:

$$\frac{\partial s(\mathbf{x_n})}{\partial f_1(\mathbf{x_n})} = \frac{\partial s(\mathbf{x_n})}{\partial f_2(\mathbf{x_n})} = 0,$$

which simplifies to:

$$f_2^2(\mathbf{x_n}) + f_1(\mathbf{x_n})f_2(\mathbf{x_n}) = f_1^2(\mathbf{x_n}) + f_1(\mathbf{x_n})f_2(\mathbf{x_n}) = 0.$$

This is equivalent to:

$$f_1(\mathbf{x_n}) + f_2(\mathbf{x_n}) = 0.$$

$\square$

**Theorem.** *(Soft Logit Normalization) For all $\mathbf{x_n} \notin \Omega$, the following equation holds:*

$$\frac{\partial \mathcal{L}_{SLN}}{\partial \mathbf{x_n}} = c_1 \gamma \frac{\partial \mathcal{L}_{LN}}{\partial \mathbf{x_n}} + c_2(1 - \gamma)\frac{\partial \mathcal{L}_{CE}}{\partial \mathbf{x_n}},$$

*where $c_1$ and $c_2$ are real positive numbers that depend on $\gamma, \tau, \mathbf{f}(\mathbf{x_n})$.*

*Proof.* Define the following terms:

$$s(\mathbf{x_n}) = [f_1(\mathbf{x_n}) - f_2(\mathbf{x_n})]/(\tau \|\mathbf{f}(\mathbf{x_n})\|_2^\gamma),$$

$$p_1(\mathbf{x_n}) = \frac{e^{f_1(\mathbf{x_n})/(\tau\|\mathbf{f}(\mathbf{x_n})\|_2^\gamma)}}{e^{f_1(\mathbf{x_n})/(\tau\|\mathbf{f}(\mathbf{x_n})\|_2^\gamma)} + e^{f_2(\mathbf{x_n})/(\tau\|\mathbf{f}(\mathbf{x_n})\|_2^\gamma)}},$$

$$p_2(\mathbf{x_n}) = \frac{e^{f_2(\mathbf{x_n})/(\tau\|\mathbf{f}(\mathbf{x_n})\|_2^\gamma)}}{e^{f_1(\mathbf{x_n})/(\tau\|\mathbf{f}(\mathbf{x_n})\|_2^\gamma)} + e^{f_2(\mathbf{x_n})/(\tau\|\mathbf{f}(\mathbf{x_n})\|_2^\gamma)}}.$$

The SLN loss can then be expressed as:

$$\mathcal{L}_{SLN}(\mathbf{x_n}) = \log(1 + e^{-s(\mathbf{x_n})}),$$

where we take $n \in C_1$ as an example (the case for $n \in C_2$ is analogous). Using the chain rule of differentiation, we have:

$$\frac{\partial \mathcal{L}_{SLN}}{\partial \mathbf{x_n}} = -p_2(\mathbf{x_n})(\frac{\partial s(\mathbf{x_n})}{\partial f_1(\mathbf{x_n})} \cdot \mathbf{w_1} + \frac{\partial s(\mathbf{x_n})}{\partial f_2(\mathbf{x_n})} \cdot \mathbf{w_2}).$$

After computation, the partial derivatives of $s(\mathbf{x_n})$ with respect to $f_1(\mathbf{x_n})$ and $f_2(\mathbf{x_n})$ are given by:

$$\frac{\partial s(\mathbf{x_n})}{\partial f_1(\mathbf{x_n})} = \frac{1}{\tau}[(1 - \gamma) \cdot \frac{1}{\|\mathbf{f}(\mathbf{x_n})\|_2^\gamma} + \gamma \cdot \frac{f_2^2(\mathbf{x_n}) + f_1(\mathbf{x_n})f_2(\mathbf{x_n})}{\|\mathbf{f}(\mathbf{x_n})\|_2^{\gamma+2}}],$$

$$\frac{\partial s(\mathbf{x_n})}{\partial f_2(\mathbf{x_n})} = -\frac{1}{\tau}[(1 - \gamma) \cdot \frac{1}{\|\mathbf{f}(\mathbf{x_n})\|_2^\gamma} + \gamma \cdot \frac{f_1^2(\mathbf{x_n}) + f_1(\mathbf{x_n})f_2(\mathbf{x_n})}{\|\mathbf{f}(\mathbf{x_n})\|_2^{\gamma+2}}].$$

Next, we define the following terms:

$$p_{2,CE}(\mathbf{x_n}) = \frac{e^{f_2(\mathbf{x_n})}}{e^{f_1(\mathbf{x_n})} + e^{f_2(\mathbf{x_n})}},$$

$$p_{2,LN}(\mathbf{x_n}) = \frac{e^{f_2(\mathbf{x_n})/(\tau\|\mathbf{f}(\mathbf{x_n})\|_2)}}{e^{f_1(\mathbf{x_n})/(\tau\|\mathbf{f}(\mathbf{x_n})\|_2)} + e^{f_2(\mathbf{x_n})/(\tau\|\mathbf{f}(\mathbf{x_n})\|_2)}},$$

$$c_1 = \frac{p_2(\mathbf{x_n})}{p_{2,LN}(\mathbf{x_n})\|\mathbf{f}(\mathbf{x_n})\|_2^{\gamma-1}},$$

$$c_2 = \frac{p_2(\mathbf{x_n})}{\tau p_{2,CE}(\mathbf{x_n})\|\mathbf{f}(\mathbf{x_n})\|_2^\gamma},$$

With these definitions, we can express the gradient of $\mathcal{L}_{SLN}(\mathbf{x_n})$ as:

$$\frac{\partial \mathcal{L}_{SLN}}{\partial \mathbf{x_n}} = -p_2(\mathbf{x_n})(\frac{\partial s(\mathbf{x_n})}{\partial f_1(\mathbf{x_n})} \cdot \mathbf{w_1} + \frac{\partial s(\mathbf{x_n})}{\partial f_2(\mathbf{x_n})} \cdot \mathbf{w_2})$$

$$= c_1\gamma \cdot [-p_{2,LN}(\mathbf{x_n})(\frac{f_2^2(\mathbf{x_n}) + f_1(\mathbf{x_n})f_2(\mathbf{x_n})}{\tau\|\mathbf{f}(\mathbf{x_n})\|_2^3} \cdot \mathbf{w_1} - \frac{f_1^2(\mathbf{x_n}) + f_1(\mathbf{x_n})f_2(\mathbf{x_n})}{\tau\|\mathbf{f}(\mathbf{x_n})\|_2^3} \cdot \mathbf{w_2})]$$

$$+ c_2(1 - \gamma) \cdot [p_{2,CE}(\mathbf{x_n})(\mathbf{w_2} - \mathbf{w_1})]$$

$$= c_1\gamma\frac{\partial \mathcal{L}_{LN}}{\partial \mathbf{x_n}} + c_2(1 - \gamma)\frac{\partial \mathcal{L}_{CE}}{\partial \mathbf{x_n}}.$$

$$\square$$

## C.2 MULTI-CLASS CASE

Assume there are $k$ classes, then we can prove the following results.

**Proposition.** *(Cross-Entropy) The gradients of the data points are given by:*

$$\frac{\partial \mathcal{L}_{CE}}{\partial \mathbf{x_n}} = - \sum_{i \in [k] \setminus y_n} \frac{e^{f_i(\mathbf{x_n})}}{\sum_{j=1}^k e^{f_j(\mathbf{x_n})}}(\mathbf{w_{y_n}} - \mathbf{w_i}).$$

*Proof.* The cross-entropy loss can be expressed as:

$$\mathcal{L}_{CE}(\mathbf{x_n}) = \log(1 + \sum_{i \in [k] \setminus y_n} e^{f_i(\mathbf{x_n}) - f_{y_n}(\mathbf{x_n})}),$$

Using the definitions $f_i(\mathbf{x_n}) = \mathbf{w_i} \cdot \mathbf{x_n} + b_i$, the result follows directly. $\square$

**Proposition.** *($\gamma = 1$, termed as Logit Normalization) Let $\Omega = \{\mathbf{x} \in \mathbb{R}^d \mid f_i(\mathbf{x}) = \mathbf{w_i} \cdot \mathbf{x} + b_i = 0, \forall i \in [k]\}$. Then, for all $\mathbf{x_n} \notin \Omega$ and $\mathbf{x} \in \Omega$, the following equation holds:*

$$\langle\frac{\partial \mathcal{L}_{LN}}{\partial \mathbf{x_n}}, \mathbf{x_n} - \mathbf{x}\rangle = 0.$$

*Proof.* Define the following terms:

$$s_i(\mathbf{x_n}) = [f_{y_n}(\mathbf{x_n}) - f_i(\mathbf{x_n})]/(\tau \|\mathbf{f}(\mathbf{x_n})\|_2), \ i \in [k] \setminus y_n,$$

$$p_i(\mathbf{x_n}) = \frac{e^{f_i(\mathbf{x_n})/(\tau \|\mathbf{f}(\mathbf{x_n})\|_2)}}{\sum_{j=1}^{k} e^{f_j(\mathbf{x_n})/(\tau \|\mathbf{f}(\mathbf{x_n})\|_2)}}, \ i \in [k],$$

The logit normalization loss can then be expressed as follows:

$$\mathcal{L}_{LN}(\mathbf{x_n}) = \log(1 + \sum_{i \in [k] \setminus y_n} e^{-s_i(\mathbf{x_n})}).$$

Using the chain rule of differentiation, we have:

$$\frac{\partial \mathcal{L}_{LN}}{\partial \mathbf{x_n}} = - \sum_{i \in [k] \setminus y_n} p_i(\mathbf{x_n}) (\frac{\partial s_i(\mathbf{x_n})}{\partial f_{y_n}(\mathbf{x_n})} \cdot \mathbf{w_{y_n}} + \frac{\partial s_i(\mathbf{x_n})}{\partial f_i(\mathbf{x_n})} \cdot \mathbf{w_i} + \sum_{j \in [k] \setminus \{y_n, i\}} \frac{\partial s_i(\mathbf{x_n})}{\partial f_j(\mathbf{x_n})} \cdot \mathbf{w_j}). \quad (5)$$

After computation, the partial derivatives of $s_i(\mathbf{x_n})$ with respect to $f_{y_n}(\mathbf{x_n})$, $f_i(\mathbf{x_n})$ and $f_j(\mathbf{x_n})$ are given by:

$$\frac{\partial s_i(\mathbf{x_n})}{\partial f_{y_n}(\mathbf{x_n})} = \frac{\sum_{l \in [k] \setminus y_n} f_l^2(\mathbf{x_n}) + f_i(\mathbf{x_n}) f_{y_n}(\mathbf{x_n})}{\tau \|\mathbf{f}(\mathbf{x_n})\|_2^3},$$

$$\frac{\partial s_i(\mathbf{x_n})}{\partial f_i(\mathbf{x_n})} = - \frac{\sum_{l \in [k] \setminus i} f_l^2(\mathbf{x_n}) + f_i(\mathbf{x_n}) f_{y_n}(\mathbf{x_n})}{\tau \|\mathbf{f}(\mathbf{x_n})\|_2^3},$$

$$\frac{\partial s_i(\mathbf{x_n})}{\partial f_j(\mathbf{x_n})} = - \frac{[f_{y_n}(\mathbf{x_n}) - f_i(\mathbf{x_n})] f_j(\mathbf{x_n})}{\tau \|\mathbf{f}(\mathbf{x_n})\|_2^3}, j \in [k] \setminus \{y_n, i\}.$$

Observing these expressions, we find that:

$$\sum_{l=1}^{k} \frac{\partial s_i(\mathbf{x_n})}{\partial f_l(\mathbf{x_n})} f_l(\mathbf{x_n}) = 0.$$

Now, consider the inner product $\langle \frac{\partial \mathcal{L}_{LN}}{\partial \mathbf{x_n}}, \mathbf{x_n} \rangle$:

$$\langle \frac{\partial \mathcal{L}_{LN}}{\partial \mathbf{x_n}}, \mathbf{x_n} \rangle = - \sum_{i \in [k] \setminus y_n} p_i(\mathbf{x_n}) (\sum_{l=1}^{k} \frac{\partial s_i(\mathbf{x_n})}{\partial f_l(\mathbf{x_n})} \cdot \langle \mathbf{w_l}, \mathbf{x_n} \rangle)$$

$$= - \sum_{i \in [k] \setminus y_n} p_i(\mathbf{x_n}) [\sum_{l=1}^{k} \frac{\partial s_i(\mathbf{x_n})}{\partial f_l(\mathbf{x_n})} \cdot (f_l(\mathbf{x_n}) - b_l))]$$

$$= - \sum_{i \in [k] \setminus y_n} p_i(\mathbf{x_n}) [\sum_{l=1}^{k} \frac{\partial s_i(\mathbf{x_n})}{\partial f_l(\mathbf{x_n})} \cdot (-b_l))].$$

And for any $\mathbf{x} \in \Omega$, we find:

$$\langle \frac{\partial \mathcal{L}_{LN}}{\partial \mathbf{x_n}}, \mathbf{x} \rangle = - \sum_{i \in [k] \setminus y_n} p_i(\mathbf{x_n}) (\sum_{l=1}^{k} \frac{\partial s_i(\mathbf{x_n})}{\partial f_l(\mathbf{x_n})} \cdot \langle \mathbf{w_l}, \mathbf{x} \rangle)$$

$$= - \sum_{i \in [k] \setminus y_n} p_i(\mathbf{x_n}) [\sum_{l=1}^{k} \frac{\partial s_i(\mathbf{x_n})}{\partial f_l(\mathbf{x_n})} \cdot (-b_l)]$$

Thus, we conclude that:

$$\langle \frac{\partial \mathcal{L}_{LN}}{\partial \mathbf{x_n}}, \mathbf{x_n} \rangle = \langle \frac{\partial \mathcal{L}_{LN}}{\partial \mathbf{x_n}}, \mathbf{x} \rangle.$$

Furthermore, since $p_i(\mathbf{x_n}) > 0$ and $\mathbf{w_1}, \mathbf{w_2}, \dots, w_k$ are linear independent, non-zero vectors, $\frac{\partial \mathcal{L}_{LN}}{\partial \mathbf{x_n}} = 0$ is equivalent to:

$$\sum_{i \in [k] \setminus y_n} p_i(\mathbf{x_n}) [\sum_{l \in [k] \setminus y_n} f_l^2(\mathbf{x_n}) + f_i(\mathbf{x_n}) f_{y_n}(\mathbf{x_n})] = 0$$

for all $i \in [k] \setminus y_n$ and $j \in [k] \setminus \{y_n, i\}$. $\qquad \square$

Notably, from the above proof we can find that if we define:

$$\frac{\partial \mathcal{L}_{LN,i}}{\partial \mathbf{x_n}} = -p_i(\mathbf{x_n})(\frac{\partial s_i(\mathbf{x_n})}{\partial f_{y_n}(\mathbf{x_n})} \cdot \mathbf{w_{y_n}} + \frac{\partial s_i(\mathbf{x_n})}{\partial f_i(\mathbf{x_n})} \cdot \mathbf{w_i} + \sum_{j \in [k] \setminus \{y_n, i\}} \frac{\partial s_i(\mathbf{x_n})}{\partial f_j(\mathbf{x_n})} \cdot \mathbf{w_j}), \ i \in [k] \setminus y_n,$$

i.e., the terms depicted in the right hand side of equation (5) in the above proof, they satisfy:

$$\frac{\partial \mathcal{L}_{LN}}{\partial \mathbf{x_n}} = \sum_{i \in [k] \setminus y_n} \frac{\partial \mathcal{L}_{LN,i}}{\partial \mathbf{x_n}},$$

then for all $\mathbf{x_n} \notin \Omega$ and $\mathbf{x} \in \Omega$, the following equation holds:

$$\langle \frac{\partial \mathcal{L}_{LN,i}}{\partial \mathbf{x_n}}, \mathbf{x_n} - \mathbf{x} \rangle = 0.$$

And if we assume that $\mathbf{w_1}, \mathbf{w_2}, \ldots, \mathbf{w_k}$ are linearly independent, non-zero vectors, then $\frac{\partial \mathcal{L}_{LN,i}}{\partial \mathbf{x_n}} = 0$ if and only if $\frac{\partial s_i(\mathbf{x_n})}{\partial f_j(\mathbf{x_n})} = 0, \ \forall i \in [k]$, which is equivalent to:

$$\sum_{l \in [k] \setminus y_n} f_l^2(\mathbf{x_n}) + f_i(\mathbf{x_n}) f_{y_n}(\mathbf{x_n}) = 0,$$

$$\sum_{l \in [k] \setminus i} f_l^2(\mathbf{x_n}) + f_i(\mathbf{x_n}) f_{y_n}(\mathbf{x_n}) = 0,$$

$$[f_{y_n}(\mathbf{x_n}) - f_i(\mathbf{x_n})] f_j(\mathbf{x_n}) = 0, \ \forall j \in [k] \setminus \{y_n, i\}.$$

From this, it follows that the above equations are equivalent to $f_j(\mathbf{x_n}) = 0, \ \forall j \in [k] \setminus \{y_n, i\}$ and $f_i(\mathbf{x_n}) + f_{y_n}(\mathbf{x_n}) = 0$. These results indicate that each $\frac{\partial \mathcal{L}_{LN,i}}{\partial \mathbf{x_n}}$ corresponds to a compression direction, similar to the discussion in proposition 2.

**Theorem.** *(Soft Logit Normalization) For all $\mathbf{x_n} \notin \Omega$, the following equation holds:*

$$\frac{\partial \mathcal{L}_{SLN}}{\partial \mathbf{x_n}} = \sum_{i \in [k] \setminus y_n} [c_{i,1} \gamma \frac{\partial \mathcal{L}_{LN,i}}{\partial \mathbf{x_n}} + c_{i,2}(1 - \gamma) \frac{\partial \mathcal{L}_{CE,i}}{\partial \mathbf{x_n}}],$$

*where*

$$\frac{\partial \mathcal{L}_{CE,i}}{\partial \mathbf{x_n}} = -\frac{e^{f_i(\mathbf{x_n})}}{\sum_{j=1}^k e^{f_j(\mathbf{x_n})}}(\mathbf{w_{y_n}} - \mathbf{w_i}),$$

*and $c_{i,1}$ , $c_{i,2}$ are real positive numbers that depend on $\gamma, \tau, \mathbf{f}(\mathbf{x_n})$.*

*Proof.* Define:
$$s_i(\mathbf{x_n}) = [f_{y_n}(\mathbf{x_n}) - f_i(\mathbf{x_n})]/(\tau \|\mathbf{f}(\mathbf{x_n})\|_2^\gamma), \ i \in [k] \setminus y_n,$$

$$p_i(\mathbf{x_n}) = \frac{e^{f_i(\mathbf{x_n})/(\tau \|\mathbf{f}(\mathbf{x_n})\|_2^\gamma)}}{\sum_{j=1}^k e^{f_j(\mathbf{x_n})/(\tau \|\mathbf{f}(\mathbf{x_n})\|_2^\gamma)}}, \ i \in [k],$$

The SLN loss can then be expressed as follows:

$$\mathcal{L}_{LN}(\mathbf{x_n}) = \log(1 + \sum_{i \in [k] \setminus y_n} e^{-s_i(\mathbf{x_n})}).$$

Using the chain rule of differentiation, we have:

$$\frac{\partial \mathcal{L}_{SLN}}{\partial \mathbf{x_n}} = - \sum_{i \in [k] \setminus y_n} p_i(\mathbf{x_n})(\frac{\partial s_i(\mathbf{x_n})}{\partial f_{y_n}(\mathbf{x_n})} \cdot \mathbf{w_{y_n}} + \frac{\partial s_i(\mathbf{x_n})}{\partial f_i(\mathbf{x_n})} \cdot \mathbf{w_i} + \sum_{j \in [k] \setminus \{y_n, i\}} \frac{\partial s_i(\mathbf{x_n})}{\partial f_j(\mathbf{x_n})} \cdot \mathbf{w_j}).$$

After computation, the partial derivatives are given by:

$$\frac{\partial s_i(\mathbf{x_n})}{\partial f_{y_n}(\mathbf{x_n})} = \frac{1}{\tau}[(1 - \gamma) \cdot \frac{1}{\|\mathbf{f}(\mathbf{x_n})\|_2^\gamma} + \gamma \cdot \frac{f_{y_n}(\mathbf{x_n}) f_i(\mathbf{x_n}) + \sum_{l \in [k] \setminus y_n} f_l^2(\mathbf{x_n})}{\|\mathbf{f}(\mathbf{x_n})\|_2^{\gamma+2}}],$$

$$\frac{\partial s_i(\mathbf{x_n})}{\partial f_i(\mathbf{x_n})} = -\frac{1}{\tau}[(1-\gamma) \cdot \frac{1}{\|\mathbf{f}(\mathbf{x_n})\|_2^\gamma} + \gamma \cdot \frac{f_{y_n}(\mathbf{x_n})f_i(\mathbf{x_n}) + \sum_{l \in [k] \setminus i} f_l^2(\mathbf{x_n})}{\|\mathbf{f}(\mathbf{x_n})\|_2^{\gamma+2}}],$$

$$\frac{\partial s_i(\mathbf{x_n})}{\partial f_j(\mathbf{x_n})} = -\frac{\gamma[f_{y_n}(\mathbf{x_n}) - f_i(\mathbf{x_n})]f_j(\mathbf{x_n})}{\tau\|\mathbf{f}(\mathbf{x_n})\|_2^{\gamma+2}}, j \in [k] \setminus \{y_n, i\}.$$

Next, we define the following terms:

$$p_{i,CE}(\mathbf{x_n}) = \frac{e^{f_i(\mathbf{x_n})}}{\sum_{j=1}^k e^{f_j(\mathbf{x_n})}},$$

$$p_{i,LN}(\mathbf{x_n}) = \frac{e^{f_i(\mathbf{x_n})/(\tau\|\mathbf{f}(\mathbf{x_n})\|_2)}}{\sum_{j=1}^k e^{f_j(\mathbf{x_n})/(\tau\|\mathbf{f}(\mathbf{x_n})\|_2)}},$$

$$c_{i,1} = \frac{p_i(\mathbf{x_n})}{p_{i,LN}(\mathbf{x_n})\|\mathbf{f}(\mathbf{x_n})\|_2^{\gamma-1}},$$

$$c_{i,2} = \frac{p_i(\mathbf{x_n})}{\tau p_{i,CE}(\mathbf{x_n})\|\mathbf{f}(\mathbf{x_n})\|_2^\gamma},$$

then the result follows directly:

$$\frac{\partial \mathcal{L}_{SLN}}{\partial \mathbf{x_n}} = \sum_{i \in [k] \setminus y_n} [c_{i,1}\gamma\frac{\partial \mathcal{L}_{LN,i}}{\partial \mathbf{x_n}} + c_{i,2}(1-\gamma)\frac{\partial \mathcal{L}_{CE,i}}{\partial \mathbf{x_n}}].$$

$\square$

## C.3 RELATION TO LARGE NN TRAINING

In fact, our theoretical analysis remains relevant for understanding the training dynamics of large neural networks (NNs). Specifically, the analysis focuses on trainable data points, which correspond to the evolution of penultimate-layer representations in a large NN. In practice, such a network can often be expressed as

$$f(\mathbf{x}) = \text{Linear}(\mathbf{y}(\mathbf{x}; \theta)),$$

where the linear layer represents the output head and $y(x; \theta)$ denotes the penultimate-layer representations with $\theta$ as the network parameters vector. By applying a Taylor expansion, the gradient descent dynamics of the penultimate-layer representations can be written as

$$\theta^{(t)} = \theta^{(t-1)} - \eta\frac{\partial \mathcal{L}}{\partial \theta},$$

$$\frac{\partial \mathcal{L}}{\partial \theta} = \frac{\partial \mathcal{L}}{\partial \mathbf{y}}\frac{\partial \mathbf{y}}{\partial \theta} = J_\theta^T(\mathbf{y}) \cdot \frac{\partial \mathcal{L}}{\partial \mathbf{y}},$$

$$\begin{aligned}
\mathbf{y}^{(t)} &= \mathbf{y}(\mathbf{x}; \theta^{(t)}) \\
&= \mathbf{y}(\mathbf{x}; \theta^{(t-1)} - \eta\frac{\partial \mathcal{L}}{\partial \theta}) \\
&= \mathbf{y}(\mathbf{x}; \theta^{(t-1)}) - \eta J_\theta(\mathbf{y}) \cdot \frac{\partial \mathcal{L}}{\partial \theta} + \text{h.o.t.} \\
&= \mathbf{y}^{(t-1)} - \eta J_\theta(\mathbf{y}) \cdot J_\theta^T(\mathbf{y}) \cdot \frac{\partial \mathcal{L}}{\partial \mathbf{y}} + \text{h.o.t.},
\end{aligned}$$

where $\eta$ is the learning rate, $\mathcal{L}$ is the loss function, and $J_\theta(\mathbf{y})$ is the Jacobian matrix.

For our analyzed linear model,

$$f(\hat{\mathbf{y}}) = \text{Linear}(\hat{\mathbf{y}}),$$

the training dynamics simplify to

$$\hat{\mathbf{y}}^{(t)} = \hat{\mathbf{y}}^{(t-1)} - \eta\frac{\partial \mathcal{L}}{\partial \hat{\mathbf{y}}}.$$

The key difference between the simplified model and the penultimate-layer dynamics of a large NN is the additional multiplication by $J_\theta(\mathbf{y}) \cdot J_\theta^T(\mathbf{y})$, which is a positive semi-definite matrix. Since

$$\left\langle \frac{\partial \mathcal{L}}{\partial \mathbf{y}}, J_\theta(\mathbf{y}) \cdot J_\theta^T(\mathbf{y}) \cdot \frac{\partial \mathcal{L}}{\partial \mathbf{y}} \right\rangle \geq 0,$$

the resulting direction change is less than $90°$. This implies that our theoretical analysis of gradient directions, including the compression effect, can be partially generalized to large NNs.

In summary, our theoretical analysis provides valuable insights into the training dynamics of penultimate-layer representations in large NNs, while we acknowledge that a more precise characterization of nonlinear architectures would require accurate estimation of the Jacobian matrix.

