# OpenReview forum: "From Minor Adjustment to Major Gains: Soft Logit Normalization Loss Enhances Representations and Generalization"
_ICLR.cc/2026/Conference — Submitted to ICLR 2026_

### Official Review · Reviewer_5Byj · 2025-10-16

**Soundness:** 3
**Presentation:** 3
**Contribution:** 3
**Rating:** 6
**Confidence:** 2

**Summary:**

This paper proposes a new loss, named SLN, which introduces two new hyperparameters compared to the naive logit normalization loss.
The authors show that SLN can outperform other losses, such as the cross-entropy loss, on several tasks including image and text classification.
The authors also provide a theoretical analysis of SLN and show that it can achieve improved representation separability.
The paper is well-written and easy to follow.

**Strengths:**

- The method is simple and easy to implement.
- The experiments are comprehensive, covering various datasets and tasks, and the results are convincing.
- The theoretical analysis is sound and provides insights into the benefits of SLN.
- The paper is well-structured and clearly presents the motivation, method, experiments, and analysis.

**Weaknesses:**

- ``Knowledge distillation'' is mentioned several times in the paper, but its relation to the method is somewhat weak.
- The design motivation is unclear. For example, "While training with CE loss, we find a consistent positive correlation between test accuracy and logit difference when the logit vector is normalized by its powered L2-norm". This motivation is not very convincing for why it does not directly use naive logit normalization loss.

**Questions:**

The proposed method is orthogonal to other techniques such as data augmentation; more experiments combining SLN with such techniques would be valuable to verify its effectiveness against stronger baselines. The same applies to knowledge distillation: whether SLN can be combined with distillation techniques and whether it can further improve its performance are interesting questions.
I think these experiments are important to verify the scalability of the method.

---

> ### Author Response · Authors · 2025-11-25
>
> - Weaknesses 1: We thank the reviewer for pointing out the unclear connection between knowledge distillation (KD) and our SLN loss, and we appreciate the opportunity to clarify this.
>
>   As discussed in the section of RELATED WORK, relation-based KD methods aim to transfer the relationships among representations from a larger model to a smaller one. In our observations, larger models tend to exhibit larger logit differences, which correlates with improved generalization. Our SLN loss is designed to enable small models to increase their logit differences, thereby improving their generalization performance.
>
>   In this sense, SLN can be interpreted as a form of relation-based KD, where the key contribution lies in enabling small models (e.g., BERT-Base) to achieve performance comparable to models roughly three times larger (e.g., BERT-Large). This performance gain is on par with that obtained by advanced KD techniques, highlighting SLN’s practical utility. We will incorporate this discussion into the revised paper.
> - Weaknesses 2: We thank the reviewer for the thoughtful comment regarding our design motivation and appreciate the opportunity to clarify this point.
>
>   Our empirical observations consistently show a positive correlation between logit difference and test accuracy, suggesting that increasing logit difference can improve generalization. This motivates the exploration of losses that explicitly encourage larger logit differences.
>
>   However, prior work [1] showed that plain logit normalization (LN, $\gamma = 1$) does not lead to noticeable accuracy gains. Based on our observations, this is likely because LN substantially alters the entire logit distribution. As a result, the positive correlation between logit difference and test accuracy no longer holds under LN.
>
>   Motivated by this limitation, we introduce soft logit normalization (SLN), which incorporates a tunable exponent $\gamma$ to preserve the overall logit distribution while enlarging logit differences. And we find this modification enables us to obtain substantial gains in generalization. Our theoretical analysis further provides insights explaining why SLN loss can outperform both CE and LN losses. We will revise Figure 1 in the updated manuscript to make these points clearer.
>
>   [1] Hongxin Wei et al., Mitigating Neural Network Overconfidence with Logit Normalization, ICML 2022.
> - Question 1: We thank the reviewer for the valuable suggestion to assess the scalability of SLN by combining it with data augmentation and knowledge distillation (KD). We have already evaluated SLN with several data augmentations such as RandAugment in the main text (Table 6), and here we provide additional experiments focusing on Mixup, CutMix, and KD.
>
>   **Data augmentation**: We conducted experiments combining SLN with Mixup and CutMix on ImageNet-1K using ResNet-50. The results are shown below:
>
>   | CE | SLN | CE + Mixup + Cutmix | SLN + Mixup + Cutmix  |
>   |:-------:|:-------:|:-------:|:-------:|
>   | 76.6 | 78.0 | 78.2 | 78.6 |
>
>   These results show that combining SLN with mixed-sample augmentations such as Mixup or CutMix yields only limited further improvements. This observation aligns with their underlying mechanisms: SLN with one-hot labels strengthens class separability in the representation space, while augmentations like CutMix explicitly mix images and labels, smoothing boundaries between classes (e.g., a mixture of “cat” and “dog”). Under such mixed-label regimes, the separability-enhancing advantage of SLN naturally diminishes, which may explain the reduced synergy.
>
>   In summary, SLN and Mixup/CutMix partially target similar effects through different means. SLN can therefore serve as an alternative to these augmentations—achieving competitive gains without the computational overhead of generating mixed samples. We will incorporate this discussion into the revised paper.
>
>   **Knowledge distillation**: We further investigated whether SLN can enhance knowledge distillation. On CIFAR-10, we trained a student model (ResNet-20) using a teacher model (ResNet-56) trained with either CE or SLN. The results are summarized below:
>
>   | KD + CE ResNet-56 (93.07%)| KD + SLN ResNet-56 (93.95%) |
>   |:-------:|:-------:|
>   | 92.07% | 92.75% |
>
>   KD + SLN consistently outperforms KD + CE, suggesting that SLN-trained teachers produce enhanced representations, which in turn lead to better student performance. This aligns with our theoretical analysis that SLN enhances the structure of learned representations. We will incorporate this discussion into the revised paper.

---

> > ### Comment · Reviewer_5Byj · 2025-11-26
> > **Response to Authors**
> >
> > Thanks for your clarification. I appreciate your detailed explanation and your response has addressed my concerns.

---

> > > ### Author Response · Authors · 2025-11-27
> > >
> > > We are very pleased to have addressed your questions regarding our work, and we sincerely appreciate your recognition of our efforts. We hope that this work can contribute to the AI community in understanding training dynamics and guiding loss function selection.

---

### Official Review · Reviewer_5rkk · 2025-10-29

**Soundness:** 2
**Presentation:** 2
**Contribution:** 1
**Rating:** 2
**Confidence:** 5

**Summary:**

This paper proposes Soft Logit Normalization (SLN) loss, which normalizes logits by their powered L2-norm (||f(x)||₂^γ) before applying softmax, where γ is a tunable hyperparameter. The key contribution is adding this power parameter γ to the existing Logit Normalization (LN) loss from Wei et al. (2022), which corresponds to γ=1. Through experiments on CIFAR-10/100, ImageNet-1K, and GLUE benchmarks, the authors show that γ≈0.7 provides improved generalization compared to standard cross-entropy (CE) loss and LN loss. They provide theoretical analysis showing that SLN's gradient is a linear combination of LN and CE gradients, and claim this leads to better representation separability in penultimate layers.

**Strengths:**

* The paper provides comprehensive empirical evaluation across multiple domains including both vision and language tasks, testing on datasets ranging from CIFAR to ImageNet and from BERT-BASE to BERT-LARGE, which demonstrates reasonable consistency of the approach.

* The presentation is clear and well-organized, with a logical progression from the initial motivation in Figure 1 through the method description to the experimental results.

* The proposed method is practically simple, requiring only a single-line code change and introducing minimal computational overhead, which makes it straightforward to adopt in existing codebases.

* The authors provide detailed implementation information in the appendices and commit to releasing their code, which supports reproducibility.

**Weaknesses:**

* The contribution appears to be relatively incremental, as the main novelty is adding an exponent γ to the existing Logit Normalization loss from Wei et al. (2022). The authors acknowledge on page 3 that "when γ=1, the SLN loss reduces to the logit normalization loss." This raises questions about whether the addition of a single hyperparameter constitutes a sufficient contribution for a full conference paper.

* The empirical improvements are marginal and could likely be achieved through various other standard training techniques. The typical gains of 0.5-2% are comparable to what one might obtain from adjusting learning rate schedules, modifying data augmentation strategies, or tuning weight decay.

* The theoretical analysis, while mathematically correct, may not provide substantial new insights. Theorem 1 shows that the SLN gradient is a weighted combination of LN and CE gradients, which follows naturally from the chain rule given the form of the loss. Since SLN interpolates between CE (γ→0) and LN (γ=1) by design, this result seems somewhat expected. The 2D visualizations in Figures 2-5 confirm this interpolation behavior. It would strengthen the paper to provide more surprising theoretical results, such as convergence rate bounds, generalization guarantees, or analysis explaining why γ≈0.7 should be optimal.

*  The paper lacks evaluation on out-of-distribution (OOD) detection tasks, which is notably absent given that the original Logit Normalization loss from Wei et al. (2022) was specifically designed for OOD detection and showed substantial improvements on that task. Since the gains on standard in-distribution classification appear marginal, it would be valuable to assess whether SLN provides benefits on OOD detection or other tasks where normalization-based losses have shown stronger performance.

**Questions:**

* Beyond empirical grid search, is there any theoretical or intuitive reason why γ≈0.7 should be optimal? Does this value relate to specific properties of the datasets or architectures?

* How does SLN perform on out-of-distribution detection tasks? Since the original LogitNorm was designed for OOD detection and showed strong results there, it would be valuable to see if the powered normalization (γ≠1) provides additional benefits on those tasks.

---

> ### Author Response · Authors · 2025-11-25
>
> - Weaknesses 1: We thank the reviewer for highlighting the relationship between SLN and other methods. We appreciate the opportunity to further clarify our contributions.
>
>   Our empirical results demonstrate that this adjustment leads to significant performance improvements (e.g. enabling small models to match the performance of models with approximately three times more parameters) across multiple benchmarks (e.g. CIFAR-10, ImageNet-1K and General Language Understanding Evaluation) and architectures (e.g. ResNet, Transformer). This highlights the practical value of SLN.
>
>   Beyond empirical gains, our work provides theoretical insights of the training dynamics for both CE, LN and SLN loss. Our analysis shows that SLN operates in a distinct dynamical regime—characterized by the lowest intra/inter-class ratio—rather than serving as a simple interpolation between CE and LN (as illustrated in Figure 6 of the main text). This sheds light on why SLN achieves stronger generalization.
>
>   We hope these clarifications help to highlight both practical and theoretical importance of our work.
> - Weaknesses 2: To ensure a fair comparison, all results reported in the main paper follow standard evaluation protocols widely used in the AI community. Specifically, we conduct extensive experiments and report the best performance for each model–loss combination. For example, on language tasks, we performed a grid search for learning rates (e.g., {2e-5, 3e-5, 4e-5, 5e-5} for language tasks) and selected the best checkpoint based on validation performance over multiple random restarts. Factors such as data augmentation and weight decay were kept consistent across different loss functions (see Appendix B for full implementation details).
>
>   Given this controlled setting—where strong baselines already benefit from extensive hyperparameter tuning—our proposed SLN enables a smaller model (BERT_BASE, 110M parameters) to achieve generalization performance comparable to that of a model roughly three times larger (BERT_LARGE, 340M parameters) trained with CE.
>
>   [1] Devlin, J. et al. BERT: Pre-training of Deep Bidirectional Transformers for Language Understanding. NAACL-HLT, 2019.
> - Weaknesses 3 & Question 1: We thank the reviewer for their thoughtful comments and for encouraging a deeper examination of our theoretical contributions.
>
>   The insight of our theoretical contribution is: The CE and LN dynamics lead to orthogonal representation distributions (Propositions 1 and 2). This orthogonality has not been previously identified. Consequently, SLN leverages this structure to induce complementary compression effects, leading to stronger within-class compactness than either CE or LN alone, directly contributing to better generalization.
>
>   Regarding why $\gamma\approx 0.7$ is often effective: Intuitively, this value reflects the point where the contributions of these two orthogonal compression dynamics are roughly equal, maximizing their joint impact.
>
>   Crucially, we confirm that this value is indeed related to dataset properties as pointed out by the reviewer. In additional experiments on idealized 2D data, we observed that when the initial data distribution is non-isotropic (i.e., the data is mostly compressed in one direction), the optimal $\gamma$ shifts significantly (e.g., to $\gamma=0.3$). This indicates that $\gamma$ acts as a balancing coefficient that compensates for the initial geometric properties of the data. Deriving an optimal gamma from first principles, taking full account of data structure, is a highly nontrivial problem that would merit a dedicated study of its own. We consider this an interesting direction for future work rather than something that can be resolved within the present manuscript.
> - Weaknesses 4 & Question 2: Thank you for the helpful suggestion regarding evaluating SLN on OOD detection tasks. Following your comment, we train ResNet-20 using CE and SLN losses on CIFAR-10, and evaluate OOD detection performance on SVHN using the softmax confidence score, following the same protocol as the original Logit Normalization paper [2], where the result of Logit Normalization is 97.26. The area under the receiver operating characteristic curve(AUROC) results are shown below:
>
>   | CE | SLN (Ours, $\gamma=0.7$) |
>   |:-------:|:-------:|
>   | 89.70 | 96.20 |
>
>   Although SLN performs slightly below LN on this OOD detection benchmark, it still substantially outperforms the standard CE loss. This demonstrates that SLN not only enables smaller models to achieve in-distribution generalization competitive with models nearly three times larger (trained with CE), but also retains the performance advantages of normalization-based losses on OOD detection tasks.
>
>   [2] Hongxin Wei et al., Mitigating Neural Network Overconfidence with Logit Normalization, ICML 2022.

---

> > ### Comment · Reviewer_5rkk · 2025-11-25
> >
> > While the authors provide additional experiments and clarifications, the core concern remains: the empirical gains from SLN are very marginal almost across all settings, typically 0.5–2%, and could likely be matched by standard hyperparameter tuning, regularization, or modest architectural adjustments. The theoretical analysis, though correct, is also fairly straightforward, showing an interpolation between CE and LN dynamics that largely follows from the definition of the loss. Together, the modest empirical improvements and relatively simple theoretical insights make the overall contribution incremental and insufficient to justify a full conference paper. For these reasons, I retain my original score.

---

> > > ### Author Response · Authors · 2025-11-26
> > >
> > > We thank the reviewer for their prompt response to our rebuttal. We appreciate the opportunity to further clarify the contribution of our work, both empirically and theoretically.
> > >
> > > 1. Empirical performance
> > >   - Fair Comparison Strategy: Our experiments already incorporate extensive hyperparameter tuning (e.g. learning rate, batch size, weight decay) for all baselines. We report the peak performance for every model-loss-task combination. Furthermore, all auxiliary regularization techniques (e.g. dropout, stochastic depth) and model architectures were kept strictly identical to the original standards (e.g., ResNet-20 [1], ResNet-50 [2], and BERT [3]). No specific architectural adjustments were made to favor SLN.
> > >   - Quantization of 1.5% Gain: Under these controlled conditions, a consistent gain of ~1.5% was observed on ImageNet-1K and GLUE. In the context of modern deep learning benchmarks, this margin often represents the performance gap between a base model and a model with 3$\times$ the parameters (e.g., the gap between the 110M-parameters BERT-BASE and the 340M-parameters BERT-LARGE).
> > >
> > > 2. Theoretical analysis
> > >   - Main Contribution: Our main insight (Propositions 1 and 2) establishes that CE and LN dynamics drive representations toward orthogonal distributions rather than showing an interpolation between CE and LN dynamics.
> > >   - Distinct Behavior, Not Just Interpolation: Because of this orthogonality, SLN does not merely behave as an average of the two. Instead, it produces a distinct dynamical trajectory characterized by the lowest intra/inter-class ratio (as illustrated in Figure 6).
> > >   - Analogy to Conjugate Gradient: This is conceptually similar to the Conjugate Gradient (CG) method [4]. CG introduce a modification to the plain gradient to leverage the geometric properties of the optimization landscape to correct the "zigzag" inefficiency of standard Gradient Descent. Similarly, SLN leverages the orthogonal forces of CE and LN dynamics to achieve a representation structure that neither loss can achieve individually.
> > >
> > > [1] Kaiming, H. et al. Deep residual learning for image recognition. CVPR, 2016.
> > >
> > > [2] Wightman R. et al. Resnet strikes back: An improved training procedure in timm. arXiv preprint arXiv:2110.00476, 2021.
> > >
> > > [3] Devlin, J. et al. BERT: Pre-training of Deep Bidirectional Transformers for Language Understanding. NAACL-HLT, 2019.
> > >
> > > [4] Shewchuk JR. An Introduction to the Conjugate Gradient Method Without the Agonizing Pain. 1994.

---

### Official Review · Reviewer_px7S · 2025-11-01

**Soundness:** 2
**Presentation:** 3
**Contribution:** 3
**Rating:** 4
**Confidence:** 3

**Summary:**

The paper suggests a new loss function based on soft logit normalization (SLN) prior to a softmax cross-entropy loss and provides an experimental and a theoretical framework underlying its general suitability. The proposed loss function seems thereby relatively straight forward by taking the existing logit normalisation and adding one particular hyperparameter that controls the power of the logits' L-2 norm functioning as the normalisation factor. This factor is effectively set to 0.7 throughout the paper.

In particular, the SLN-based loss is claimed to lead to better class seperability independent of task and modality and should therefore be applicable across the board. In particular, small architecture versions are shown to reach similar performance to larger architecture versions. Experimental evaluation is performed on Image data (CIFAR-10, CIFAR-100 and ImageNet-1k) and text data (GLUE). Gradient training dynamics are analysed on a theoretical level for a binary classifier and data representations are compared via intra-inter ratio training dynamics and t-SNE.

**Strengths:**

- The paper suggests a simple twist to cross-entropy loss that is easy to implement and easy to follow and seems promising to outperform standard CE loss in many to most settings. If the benefits are as described that approach has a very large potential.
- The paper offers empirical and theoretical analysis underlining the effectiveness of the approach. It generally considers many different angles under which the approach is viewed.
- The paper is clear and well written.

**Weaknesses:**

- Despite the potential overall benefit, the suggested approach sits upon previous ideas and the suggested variation (the additional hyperparameter to control for the power of the norm) is only a small adjustment to previously existing ideas.
- The experimental could further showcase a holistic evaluation of SLN, for instance its interaction w.r.t. data augmentation, as described in the questions below.
- Some further display of experiments and analysis are unclear, as further described in the questions.
- Without the publication of code, which is not promised in the submission, the work will be difficult to reproduce.
- There are no sections or any discussion about the limitations/disadvantages of the approach in the text. Are we supposed to believe that SLN is a universally applicable approach that will always be preferable to any cross entropy in any setting?

**Questions:**

Figure 1: The expressiveness of this is Figure is not very clear to me. First, why doesn't it include gamma=0, which should come equal to the use of just plain cross-entropy if I am not mistaken? This is not further discussed and not shown here. Also wouldn't such a figure be more meaningful to compare between different training runs of identical models and not only across models? Also, just because the linear correlation seems to increase with increasing normalisation, why does this imply that this is also useful for training, for instance the best performing gamma values from the ones mentioned (0.6 and 0.8) show worse linear correlation than the gamma=1.

Table 2: Why is the advanced data augmentation not applied to the SLN? And why is neither applied to ResNet-101? While the results look promising, it would indeed be interesting to see the interaction of data augmentation and SLN, whether the combination is even better or worse? Per se, I don't see a reason why they couldn't be combined and given the extensive claims of superiority of SLN over CE and LN, it would be interesting to see whether performance saturates with data augmentation or whether there would still be a benefit of using SLN. At least there should also be a discussion of possible interaction, as e.g., cutmix changes the labels and thus also equations (1) and (2). Would it be possible to extend the experiments in this direction?

Table 3: Why are all individual results across tasks are outlined in the table, if they are not further discussed and their meaning is not obvious without consulting the corresponding references? If the only relevant result is the Average across all tasks (any maybe the MNLI) it seems enough to report these, otherwise it may be insightful to discuss, why SLN for BERT Base performs sometimes notably better and sometimes notably worse than CE for BERT Large for the individual tasks.

Reproducability statement: Why is the code not planned to be made publicly available? I believe this is would be very helpful and should be common practice. While the paper does include many specifics, it may not always be obvious if there are some undefined hyperparameters, implementation details or alike that prevent the paper from being reproducable. For instance, there is no specification of random seeds, which already prevents an exact reproduction of the results listed here.

Figure 7: Does this graph really include all the mean (and I assume standard deviation or alike?) across all three datasets? This seems very unlikely givent that the performance across the three datasets varies from around 70% to 90%. Given the accuracy it seems rather to be Imagenet. If so, why is the best performance over 80% while best in Table 5 is 78%? In any case, it would be great to have this figure split by dataset to show individual results per dataset. In any case, it also not clear how many runs (I assume with different random seeds?) are performed to obtain the results both in Figure 7 and Table 5.

Why are no gamma values higher than 1 discussed? Do these have a (non-)obvious disadvantage?

---

> ### Author Response · Authors · 2025-11-25
>
> - Weaknesses 1: We thank the reviewer for highlighting the relationship between SLN and other methods. We appreciate the opportunity to further clarify our contributions.
>
>   Our empirical results demonstrate that this adjustment leads to significant performance improvements (e.g. enabling small models to match the performance of models with approximately three times more parameters) across multiple benchmarks (e.g. CIFAR-10, ImageNet-1K and General Language Understanding Evaluation) and architectures (e.g. ResNet, Transformer). This highlights the practical value of SLN.
>
>   Beyond empirical gains, our work provides theoretical insights of the training dynamics for both CE, LN and SLN loss. Our analysis shows that SLN operates in a distinct dynamical regime—characterized by the lowest intra/inter-class ratio—rather than serving as a simple interpolation between CE and LN (as illustrated in Figure 6 of the main text). This sheds light on why SLN achieves stronger generalization.
>
>   We hope these clarifications help to highlight both practical and theoretical importance of our work.
> - Weaknesses 2 (Table 2): We thank the reviewer for the detailed observations. Indeed, we have conducted additional experiments evaluating the interaction between SLN and data augmentation techniques. The results (ImageNet-1K, ResNet-50) are provided below:
>
>   | CE | SLN | CE + Mixup + Cutmix | SLN + Mixup + Cutmix |
>   |:-------:|:-------:|:-------:|:-------:|
>   | 76.6 | 78.0 | 78.2 | 78.6 |
>
>   These results show that combining SLN with mixed-sample augmentations such as Mixup or CutMix yields only limited further improvements. This observation aligns with their underlying mechanisms: SLN with one-hot labels strengthens class separability in the representation space, while augmentations like CutMix explicitly mix images and labels, smoothing boundaries between classes (e.g., a mixture of “cat” and “dog”). Under such mixed-label regimes, the separability-enhancing advantage of SLN naturally diminishes, which may explain the reduced synergy.
>
>   In summary, SLN and Mixup/CutMix partially target similar effects through different means. SLN can therefore serve as an alternative to these augmentations—achieving competitive gains without the computational overhead of generating mixed samples. We will incorporate this discussion into the revised paper.
> - Weaknesses 3:
> 1. Figure 1:  (a) Is $\gamma = 0$ equal to the use of just plain cross-entropy? Yes, $\gamma = 0$ corresponds exactly to plain cross-entropy. (b) Why compare across different model sizes instead of repeated runs of identical models? We have conducted multiple training runs of the same model and observed a consistent trend: larger networks exhibit larger logit differences when the logits are normalized by their powered L2 norm. We will clarify this in the revised version. (c) Why the best choice of $\gamma$ is not 1? Our experiments reveal a positive relationship between logit difference and test accuracy; thus, increasing logit difference should benefits generalization (not necessarily in a strictly linear fashion).
>
>     However, prior work [1] showed that plain logit normalization (LN, $\gamma = 1$) does not lead to noticeable accuracy gains. Based on our observations, this is likely because LN substantially alters the entire logit distribution. As a result, the positive correlation (between logit difference and test accuracy) no longer holds under LN.
>
>     Motivated by this limitation, we introduce soft logit normalization (SLN), which incorporates a tunable exponent $\gamma$ to preserve the overall logit distribution while enlarging logit differences. And we find this modification enables us to obtain substantial gains in generalization. Our theoretical analysis further provides insights explaining why SLN loss can outperform both CE and LN losses. We will revise Figure 1 in the updated manuscript to make these points clearer.
>
>      [1] Hongxin Wei et al., Mitigating Neural Network Overconfidence with Logit Normalization, ICML 2022.
> 2. Table 3: We follow the original BERT evaluation protocol and therefore report results on all tasks for completeness. The primary takeaway is the Average score, which shows that SLN enables smaller models (BERT-Base, 110M) to achieve comparable performance to much larger models (BERT-Large, 340M) trained with CE.
>
>      Following the reviewer’s suggestion, we will add task descriptions in the Appendix and expand our discussion. For example, SLN performs particularly well on CoLA, a binary single-sentence classification task, where the goal is to predict whether an English sentence is linguistically “acceptable” or not, suggesting that SLN is especially effective for these tasks.
> 3. Figure 7: Figure 7 include all the mean. For example, for $\gamma=0.7$:
>       $$(92.58+69.75+78.00)/3 = 80.11$$
>
>      Following the reviewer’s suggestion, we would provide accuracy curves separately for each dataset in the revised paper.

---

> ### Author Response · Authors · 2025-11-25
>
> - Weaknesses 4: We agree with the reviewer on the importance of reproducibility. We will release the full codebase, along with training scripts and configuration files, upon acceptance to ensure that all results in the paper can be fully reproduced.
> - Weaknesses 5: We appreciate the reviewer’s question. Based on our theoretical analysis and empirical findings, SLN is effective for most one-hot–label classification tasks, where improving class separability is beneficial. In these scenarios, SLN can indeed serve as a strong and often preferable alternative to cross-entropy.
>
>   However, for tasks not primarily driven by inter-class separability—such as next-token prediction in language modeling—it remains unclear whether SLN provides advantages over CE, and further investigation is needed. In addition, as discussed earlier, SLN does not combine effectively with certain data augmentation techniques (e.g., CutMix), which can smooth class boundaries in ways that partially counteract SLN’s effect.
>
>   We will include a dedicated section discussing these limitations in the revised version.
> - Question about $\gamma > 1$: We thank the reviewer for raising this question. We have conducted experiments with $\gamma > 1$, and the results (CIFAR-10, ResNet-20) are shown below:
>
>     | 0.0 | 0.7 | 1.0 | 1.5 | 2.0 |
>     |:-------:|:-------:|:-------:|:-------:|:-------:|
>     | 91.81$\pm$0.15\% | 92.58$\pm$0.08% | 92.05$\pm$0.10% | 91.11$\pm$0.22% | 90.78$\pm$0.25% |
>
>   As shown, performance monotonically degrades once $\gamma>1$. This behavior aligns with our theoretical analysis. According to Theorem 1 in the main paper, when $\gamma>1$ , the direction of the CE gradient component becomes opposite. Therefore, instead of encouraging inter-class separation, $\gamma>1$ induces inter-class compression, which degrades generalization performance.
>
>   For this reason, we focus on the practical range $\gamma \in [0, 1]$ in the main experiments. We will incorporate this discussion into the revised paper.

---

### Official Review · Reviewer_aqLd · 2025-11-01

**Soundness:** 3
**Presentation:** 3
**Contribution:** 3
**Rating:** 8
**Confidence:** 2

**Summary:**

The paper proposes the Soft Logit Normalization (SLN) loss, which normalizes logits by a powered L2 norm before applying softmax. This simple modification yields consistent generalization gains across vision (CIFAR, ImageNet) and language (GLUE) benchmarks, allowing small models to match much larger ones. The authors also provide theoretical analysis showing that SLN promotes more separable representations, potentially explaining its improved generalization.

**Strengths:**

- Introduces a simple yet generalizable loss (Soft Logit Normalization) that yields consistent and significant generalization gains across both vision and language tasks.

- Provides strong empirical evidence on CIFAR, ImageNet, and GLUE, showing small models can match much larger ones or distillation baselines.

- Offers a clear theoretical analysis explaining improved representation separability and connects theory with visualization experiments.

**Weaknesses:**

- Since state-of-the-art vision results now rely heavily on Vision Transformers, it would strengthen the paper to test SLN on those architectures ls to assess its generality and potential for pushing SOTA performance.
- I’m not an expert in loss design, so I’ll defer to other reviewers regarding the completeness of the baseline comparisons.

**Questions:**

The idea that the proposed loss enhances representation separability is intriguing. Conceptually this might benefit transfer learning, OOD robustness and potentially more. It would be interesting to conduct experiments in these scenarios.

---

> ### Author Response · Authors · 2025-11-25
>
> - Weaknesses 1: We thank the reviewer for raising this important point. To further evaluate the generality of SLN, we additionally tested it on Vision Transformers (ViT) pretrained on ImageNet-1K. The top-1 accuracy results are shown below:
>
>   | Model | Loss | Top-1 acc. |
>   |:-------:|:-------:|:-------:|
>   | ViT-Ti(5.7M) | CE | 73.28 |
>   | ViT-Ti(5.7M) | SLN(Ours) | 74.37 |
>   | ViT-S(22M) | CE | 74.52 |
>   | ViT-S(22M)  | SLN(Ours) | 75.66 |
>
>   These results demonstrate that SLN consistently improves performance on ViT architectures, indicating that its benefits are not limited to convolutional networks. This finding is also consistent with our results on language tasks using BERT [1] (Table 3 in the main text), further supporting the generality of SLN across architectures. Overall, these experiments suggest that SLN has strong potential for pushing state-of-the-art performance across a broad range of models. We will incorporate this result with experimental details into the revised paper.
>
>   [1] Devlin, J. et al. BERT: Pre-training of Deep Bidirectional Transformers for Language Understanding. NAACL-HLT, 2019.
> - Weaknesses 2: We appreciate the reviewer’s comment and would like to clarify how we selected the baselines. To ensure fairness, we chose baselines that are commonly used in the relevant literature, and compatible with the training paradigm and model scale used in our experiments.
>
>   Within this scope, we evaluated baselines that satisfy these criteria and demonstrated that our method consistently outperforms representative and widely adopted alternatives (e.g. Cross-Entropy loss).
> - Question 1: We thank the reviewer for the insightful suggestion. Following your recommendation, we conducted additional transfer learning experiments to assess whether the improved representation separability induced by SLN benefits downstream performance.
>
>   Specifically, we evaluated the ViT-Ti model pretrained on ImageNet-1K using either CE or SLN, and fine-tuned it on CIFAR-10. For fairness, all fine-tuning was performed using CE loss. The results are shown below:
>
>   | Model | Loss when pre-training on ImageNet-1K | Acc. on CIFAR-10 |
>   |:-------:|:-------:|:-------:|
>   | ViT-Ti(5.7M) | CE | 83.87 |
>   | ViT-Ti(5.7M) | SLN(Ours) | 84.41 |
>
>   These results indicate that SLN-trained models indeed transfer better and exhibit improved robustness on an out-of-distribution dataset. This supports the intuition that enhanced representation separability can benefit downstream tasks, and suggests that SLN may serve as an effective pretraining objective for transfer learning scenarios.

---

### Meta-Review · Area_Chair_xN4Z · 2026-01-04

**Summary:**

The paper proposes Soft Logit Normalization, a loss function modification that normalizes logits by their powered L2-norm before Softmax. The authors claim this enhances representation separability and generalization, enabling smaller models to match the performance of larger ones. The submission includes theoretical analysis suggesting SLN induces distinct training dynamics compared to CE and standard LN

**Reviewer Concerns:**

## Addressed:

- The authors provided additional experiments on ViT and transfer learning as requested.
- Clarifications on KD connections and additional KD experiments is added.
- Authors tested on SVHN (OOD), showing SLN outperforms CE but slightly lags behind the original LN

## Outstanding:
- The most critical outstanding concern is that SLN is mathematically a minor variation to the existing LN. While the authors argue theoretical orthogonality, the novelty issue may still be valid.
- A crucial one  is that SLN does not stack effectively with standard augmentations like Mixup/CutMix. The authors admit SLN provides limited gains in these regimes because mixed labels partially counteract SLN’s effect. This limits the method's use in current training pipelines where such augmentations are standard.

**Reviewer Scores:**

Reviewer aqLd: 8==> 8. Positive initially and the rebuttal responsed the requests for ViT results.

Reviewer px7S: 4 ==> 4.  The confirmation that SLN conflicts with Mixup/Cutmix  likely cannot sway this reviewer.

Reviewer 5rkk: 2 ==> 2. This reviewer keep viewing the core contribution as too incremental.

Reviewer 5Byj: 6 ==> 6. Concerns are addressed.

---

### Decision · Program_Chairs · 2026-01-26

Reject